# Plasmid fitness costs are caused by specific genetic conflicts enabling resolution by compensatory mutation

James P. J. Hall[1,2]*, Rosanna C. T. Wright[2,3], Ellie Harrison[2], Katie J. Muddiman[2], A. Jamie Wood[4,5], Steve Paterson[1], Michael A. Brockhurst[3]

**1** Department of Evolution, Ecology and Behaviour, Institute of Infection, Veterinary and Ecological Sciences, University of Liverpool, Liverpool, United Kingdom, **2** Department of Animal and Plant Sciences, University of Sheffield, Sheffield, United Kingdom, **3** Division of Evolution and Genomic Sciences, University of Manchester, Manchester, United Kingdom, **4** Department of Biology, University of York, York, United Kingdom, **5** Department of Mathematics, University of York, York, United Kingdom

* j.p.j.hall@liverpool.ac.uk

**Data Availability Statement:** Data and example analysis scripts are available at https://github.com/ jpjh/COMPMUT/ and at the University of Liverpool

## Abstract

Plasmids play an important role in bacterial genome evolution by transferring genes between lineages. Fitness costs associated with plasmid carriage are expected to be a barrier to gene exchange, but the causes of plasmid fitness costs are poorly understood. Single compensatory mutations are often sufficient to completely ameliorate plasmid fitness costs, suggesting that such costs are caused by specific genetic conflicts rather than generic properties of plasmids, such as their size, metabolic burden, or gene expression level. By combining the results of experimental evolution with genetics and transcriptomics, we show here that fitness costs of 2 divergent large plasmids in *Pseudomonas fluorescens* are caused by inducing maladaptive expression of a chromosomal tailocin toxin operon. Mutations in single genes unrelated to the toxin operon, and located on either the chromosome or the plasmid, ameliorated the disruption associated with plasmid carriage. We identify one of these compensatory loci, the chromosomal gene *PFLU4242*, as the key mediator of the fitness costs of both plasmids, with the other compensatory loci either reducing expression of this gene or mitigating its deleterious effects by up-regulating a putative plasmid-borne ParAB operon. The chromosomal mobile genetic element Tn6291, which uses plasmids for transmission, remained up-regulated even in compensated strains, suggesting that mobile genetic elements communicate through pathways independent of general physiological disruption. Plasmid fitness costs caused by specific genetic conflicts are unlikely to act as a long-term barrier to horizontal gene transfer (HGT) due to their propensity for amelioration by single compensatory mutations, helping to explain why plasmids are so common in bacterial genomes.

Datacat, doi: 10.17638/datacat.liverpool.ac.uk/1275. Short read sequences are available on NCBI-GEO short-read archive, project accession GSE151570.

**Funding:** This work was supported by funding from NERC [to M.A.B., S.P., A.J.W., J.P.J.H. NE/R008825/1; to M.A.B., A.J.W. NE/K011774/1], an ERC Consolidator Grant [to M.A.B.; 11490-COEVOCON], an Academy of Medical Sciences Springboard Award [to J.P.J.H.; SBF005\1062], and funding from the Institutional Strategic Support Fund (ISSF) awarded by Wellcome Trust via the University of Liverpool [to J.P.J.H.; 204822/Z/16/Z]. The funders had no role in study design, data collection and analysis, decision to publish, or preparation of the manuscript.

**Competing interests:** The authors have declared that no competing interests exist.

**Abbreviations:** AIC, Akaike information criterion; FDR, false discovery rate; GC content, guanine-cytosine content; GLM, generalized linear model; GmR, gentamicin resistance; GO, Gene Ontology; HGT, horizontal gene transfer; KB, King's B; LMM, linear mixed-effects model; ORF, open reading frame; RNA-seq, RNA sequencing; SmR, streptomycin resistance.

## Introduction

Plasmid-mediated horizontal gene transfer (HGT) is a powerful force in bacterial evolution. Many of the traits necessary for the ecological success of bacteria in new environments are carried between bacterial strains and species by plasmids. In hospitals, plasmids mobilise antimicrobial resistance genes between pathogens, commensals, and environmental bacteria of diverse species [1–3]. In soil, plasmids are responsible for bioremediation of contaminated sites and provide new ecological opportunities both to free-living and to plant-associated bacteria [4,5]. In infections, plasmids provide key virulence determinants necessary for colonisation and establishment and factors that assist in immune evasion and pathogen transmission [6,7]. Thus, in diverse habitats, plasmids are integral constituents of microbiomes, acting as potent drivers of bacterial evolutionary ecology, in large part through their role as vehicles of HGT [8–11].

However, there exist various features of recipient bacteria that restrict plasmid transmission and thus the flow of genetic information [12]. Some of these features impede acquisition of plasmids, such as entry exclusion systems, CRISPR, and restriction systems [13,14]. Other barriers emerge from the inability of plasmids to replicate or to segregate effectively to daughter cells. Importantly, even if a plasmid can enter, replicate, and be passed to daughter cells, its longer-term persistence is contingent upon the fitness costs that emerge from plasmid carriage [15,16].

Plasmid-imposed fitness costs are the negative impacts on reproduction and/or survival that emerge as a consequence of plasmid carriage [15]. These costs are thought to act as major barriers to plasmid maintenance—and plasmid-mediated HGT—because cells burdened by the costs of plasmid carriage are outcompeted by others that lack the plasmid and its associated costs [17]. Importantly, selection for plasmid-borne genes (e.g., by antibiotics) is insufficient to explain plasmid persistence in the long term, because those genes can become relocated to the chromosome through recombination, resulting in net fitness costs of plasmid carriage, favouring plasmid loss [17–19]. Therefore, to understand plasmid maintenance—and by extension the ability of plasmids to act as vehicles of gene exchange in bacterial communities—it is crucial to understand the mechanisms that underlie plasmid fitness costs and how these costs may be resolved.

A variety of causes of plasmid fitness costs have been proposed [15], but in most cases, the underlying molecular mechanisms remain unknown. General properties of plasmids may explain costs because cellular resources—not least nucleotides, amino acids, and the machineries of replication, transcription, and translation—must be redirected to plasmid activity, imposing a metabolic burden [16,20]. These costs would be expected to scale with plasmid size, copy number, and gene expression, being higher for plasmids that are large, multicopy, and/or have highly active promoters. Plasmid DNA composition, particularly guanine-cytosine (GC) content and bias in codon use, is another general property that can be a potential source of plasmid costs, exacerbating the inefficiencies in bacterial physiology caused by plasmid maintenance [21,22]. Alternatively, plasmid fitness costs can derive from the effects of specific plasmid genes. Some plasmid-borne genes are costly in and of themselves, for example, some efflux pumps [23] or the conjugative pilus that can disrupt the cell membrane and exposes bacteria to phage predation [24], whereas other plasmid genes elicit deleterious interactions with chromosomal genes, for example, by triggering stress responses [25] or interfering with metabolism [26,27].

Commonly, plasmids or the host bacterium undergo compensatory evolution, reducing the costs of plasmid carriage and hence the strength of purifying selection against plasmids, enabling maintenance [15,23,28–33]. Intuitively, the nature of compensatory mutations can provide an insight into the causes of plasmid fitness costs. If costs emerge from general

properties of plasmids, we might expect large-scale changes, such as deletions that reduce plasmid size, or many individual changes to plasmid codon usage. On the other hand, if costs derived from specific plasmid genes or specific plasmid–chromosomal gene interactions, we might instead expect targeted deletions, mutations, or regulatory changes affecting only those genes. While some studies of compensatory evolution do report large-scale plasmid deletions [28,34], the majority implicate one or few mutations to specific genes as responsible for ameliorating the cost of plasmid carriage [23,25,29–33]. Correspondingly, meta-analysis of fitness costs estimates found no correlation between the size of plasmids and their fitness costs [35]. The available data are therefore consistent with the hypothesis that specific genetic interactions may commonly underlie plasmid fitness costs. Identifying these interactions, and how they are resolved by compensatory evolution, is crucial for understanding the evolution and ecology of HGT. In particular, the location of compensatory evolution is predicted to have important consequences: If the compensatory mutation occurs to the plasmid, then it is propagated with onwards horizontal transmission [30,36], whereas mutations occurring on the chromosome can potentially facilitate accumulation of many different plasmids [32,37]. However, chromosomal and plasmid compensations may not both be available for some plasmid–host combinations owing to the mechanism by which plasmid costs emerge and are resolved. Although many compensatory mutations have been identified for different plasmid–bacterial pairings, the mechanistic bases of compensation and thus the nature of the resolved fitness costs, often remain unclear.

Here, we take advantage of a plasmid–host system where multiple genetic pathways of compensatory evolution exist, to investigate the mechanistic bases of plasmid fitness costs and their amelioration. We have previously described 2 pathways of chromosomal compensatory mutation in *Pseudomonas fluorescens* SBW25, targeting the *gacA/S* global regulatory system or the hypothetical protein *PFLU4242*, which individually ameliorate the fitness costs of megaplasmids pQBR103 (425 kb), pQBR57 (307 kb), or pQBR55 (157 kb) [29,38]. In this study, we describe an additional plasmid-borne compensatory locus, the lambda repressor–like protein PQBR57_0059, which ameliorates the cost of pQBR57. Using a combination of experimental evolution, transcriptomics, and genetics, we then show that specific genetic conflicts between chromosomal and plasmid genes, not plasmid size or gene expression per se, are the principal cause of plasmid fitness costs in this system and identify a single chromosomal locus that acts as the key mediator of plasmid fitness costs.

## Materials and methods

### Bacterial strains

*P. fluorescens* SBW25 was cultured in 6-ml King's B (KB) media in 30-ml glass universals ("microcosms") at 28˚C shaking at 180 rpm. Strains were labelled with either gentamicin resistance (GmR) or streptomycin resistance (SmR) by insertion of mini-Tn7 resistance cassettes into *attTn7* sites according to the protocol of Choi and Schweizer [39]. Allelic replacement of *gacA*, *gacS*, and *PFLU4242* genes was achieved using the pUIC3 suicide vector with a 2-step protocol as described previously [33]. The "evolved" pQBR57 plasmids came from strains A.01.65.G.002 (ancestral type) and B.09.65.G.029 (V100A variant; samples 1 and 49, respectively, from [40]). These plasmids were allowed to transfer into an ancestral SmR background which were used as donor strains for conjugation into the experimental lines. Plasmid pQBR103 was from [41].

Transconjugants for use in experiments were generated by mixing equal volumes of overnight cultures of donors and recipients and coculturing a 1:100 dilution in liquid KB for 24 hours before spreading on KB plates containing mercuric chloride (20 μM) and either gentamicin (30 μg/ml) or streptomycin (250 μg/ml). Candidate transconjugant colonies were restreaked on selective media and tested by PCR for the presence of the plasmid. Independent

transconjugants were used for each replicate. Conjugation rates were estimated using a similar protocol, except start and endpoint cultures were also spread on KB agar supplemented with 50 μg/ml X-gal to enumerate cfu of donors and recipients, and conjugation rate was calculated using the Simonsen and colleagues [42] method.

To express PQBR57_0059 independently of pQBR57, amplicons from the "anc" and "V100A" strains were cloned into the EcoRI/KpnI site of pUCP18 [43]. Inserts were sequenced, and 500-ng plasmid was transformed into *P. fluorescens* SBW25 GmR using the sucrose method [39]. Three independent transformants were generated (1 for each replicate). Putative pQBR57 *par* genes were cloned into pUCP18 and transformed into *P. fluorescens* SBW25 in a similar manner. pUCP18-containing lines were selected and maintained on kanamycin 50 μg/ml, and pQBR57 and pQBR103 variants were transferred to these lineages as described above. To test effects of expressing plasmid-regulated genes, candidates were cloned into the EcoRI/KpnI site of pME6032 [44] (or, for *PFLU4242*, the SacI/KpnI site), sequenced, and transformed into *P. fluorescens* as above. pME6032-containing lines were selected and maintained on 10 μg/ml (*Escherichia coli*) or 100 μg/ml (*P. fluorescens*) tetracycline, and expression was induced with 100 μM IPTG.

## Competitions

For all competitions, GmR strains were competed against SmR-*lacZ*+ [45] strains. Overnight cultures of competing strains were washed and resuspended in KB, mixed in approximately equal ratios, and used to inoculate a KB microcosm at a 1:100 dilution. Samples of the competitors were spread on KB agar supplemented with 50 μg/ml X-gal to estimate colony forming units at the beginning of the experiment and after 48 hours, and relative fitness was calculated as the ratio of Malthusian parameters $W = \ln(\text{test}_{end}/\text{test}_{start})/\ln(\text{reference}_{end}/\text{reference}_{start})$ [46]. For all competitions, the reference strain was the plasmid-free SmR-*lacZ* strain, except where mentioned in **Fig 2B** (where the reference strain was the plasmid-free GmR strain). These assays are designed to assess the success of lineages that initially start with the plasmid, rather than plasmids per se, since competitors may gain or lose the plasmid during the assay. We considered this the most appropriate measure of fitness, as we were interested in understanding the impact of plasmid carriage on a bacterial host. We also used replica and selective plating to assess plasmid loss and conjugative transfer. Segregational loss had a negligible effect on the overall proportions of plasmid bearers over the 48 hours of the competition (<0.01% of initially resistant colonies became susceptible, see also Hall and colleagues [47] and **S8 Fig**). For the highest conjugation rate conditions (into *P. fluorescens*, where the donor has both compensatory mutations), transconjugants comprised approximately 10% of the final plasmid-bearing population. Including such transconjugants in the fitness calculations (i.e., defining $W_{plasmid} = \ln(\text{plasmid-bearing}_{end}/\text{plasmid-bearing}_{start})/\ln(\text{plasmid-free}_{end}/\text{plasmid-free}_{start})$) resulted in fitness values that were slightly greater (<7% on average) than those calculated above, but did not qualitatively affect study findings.

"Medium-term" competitions in soil were performed as described in [40]. Briefly, equal ratios of plasmid bearers and plasmid-free strains were mixed and added to soil microcosms and transferred into fresh media every 4 days for 5 transfers in total. Populations were enumerated by plating or replica plating onto selective media.

## Growth curves

Growth curves of pME6032-carrying lines were established from a 1:1,000 dilution of overnight culture into 150 μl KB + tetracycline ± 100 μM IPTG. Plates with growth curve cultures were incubated under humid conditions at 28°C shaking at 180 rpm (3-mm orbital radius),

and OD600 was measured every 15 minutes on a Tecan Spark 10M plate reader. For testing plasmid-induced genes, 4 independent replicate cultures were established for each of 3 independent transformants. For testing *PFLU4242* expression, one replicate was conducted from each of 4 independent transformants/conjugations. Corrected OD600 was calculated by subtracting, from each reading, the mean absorbance from blank (no bacteria added) wells from that growth curve run.

## RNA extraction and sequencing

RNA was extracted using TRI Reagent (T9424, Sigma-Aldrich, Gillingham, Dorset). Overnight cultures of bacteria were subcultured 1:50 into 6-ml prewarmed KB microcosms and grown at 28˚C shaking at 180 rpm. When cultures reached intermediate exponential phase (OD600 approximately 0.6, as assessed on a Tecan Spark plate reader), 1-ml samples were added to 0.4 volumes of ice-cold 95% ethanol 5% phenol "stop solution" [48], mixed gently, and incubated on ice for at least 30 minutes. Each of the 3 replicates was processed separately as a block. Samples were harvested at 4,500 G for 10 minutes, resuspended in 1 ml TRI reagent, and incubated at room temperate for 5 minutes. Amylene-stabilised chloroform (400 μl, Sigma-Aldrich 34854-M) was added to each tube and incubated for 2 to 15 minutes at room temperature. Samples were then transferred to 5Prime Phase Lock Gel "Heavy" tubes (733–2478, VWR, Lutterworth, Leicestershire) and centrifuged at 17,000 G for 15 minutes at room temperature. The aqueous phase was collected into a fresh microfuge tube, and 450-μl isopropanol was added to precipitate RNA. Samples were incubated at room temperature for 30 minutes before pelleting at 17,000 G for 30 minutes. The pellet was rinsed by gently adding 1 ml of 70% ethanol and pelleting at 7,500 G for 5 minutes. The supernatant was carefully removed, and the pellet was allowed to air dry before resuspension in RNase-free water at 65˚C. To remove any residual contaminating DNA, RNA was diluted to ≤200 ng/μl and treated with TURBO DNA-free RNase-free DNase (AM1907, Invitrogen, Loughborough, Leicestershire) according to the manufacturer's instructions. Reaction products were purified with Agencourt RNAClean XP beads. Ribosomal RNA was removed using the Bacterial RiboZero rRNA depletion kit (Illumina, Cambridge, Cambridgeshire), and stranded libraries generated using NEBNext Ultra-Directional RNA library preparation kit (Illumina). Libraries were sequenced using paired-end 2 × 150 bp sequencing on 2 lanes of a HiSeq 4000. Short read sequences are available on NCBI-GEO short-read archive, project accession GSE151570.

## RNA-seq analysis

Illumina adapter sequences were trimmed from the raw FASTQ files using Cutadapt version 1.2.1, using the option -O 3. The reads were further trimmed using Sickle version 1.200 (https://github.com/najoshi/sickle), with a minimum window quality score of 20, and reads shorter than 20 bp after trimming were removed. FastQC (http://www.bioinformatics.babraham.ac.uk/projects/fastqc/) and MultiQC [49] were used to analyse the reads. Warnings arising from "Sequence content," "Sequence length," and "Sequence duplication" modules were followed up and found to be largely due to an overrepresentation of functional RNAs, namely tmRNA and RnaseP. Reads from each sample were aligned strand specifically using HISAT2 version 2.1.0 [50] to the corresponding reference sequence(s): either the *P. fluorescens* SBW25 chromosome (AM181176) alone, the chromosome with pQBR57 (LN713926), or the chromosome with pQBR103 (AM235768). The options phred33, no-spliced-alignment, new-summary, and rna-strandness RF were used, and the output was filtered using samtools to remove mappings with PHRED-scaled quality <10. The function featureCounts from the Rsubread package version 1.30.9 [51] was used to identify the reads mapping to each feature, and the table of counts was analysed in edgeR version 3.22.5 [52]. Here, we restricted differential

expression analysis to putative protein-coding genes. Expression of chromosomal genes was analysed in a set of negative binomial generalized linear models (GLMs), and all treatments were compared with the plasmid-free ancestor with a series of quasi-likelihood F tests. $p$-Values were corrected using the Benjamini–Hochberg false discovery rate (FDR) of q = 0.05.

## Statistical analyses

Relative fitness of knockout strains was analysed in a linear mixed-effects model (LMM) using the R package nlme. The data were Box-Cox transformed to meet model assumptions. A model was fitted to all interactions of plasmid and host genotype. Transconjugant replicate was included as a random effect. Model reduction was performed using stepwise Akaike information criterion (AIC) comparisons and likelihood ratio tests. Effects of mutations and of plasmid carriage were estimated using post hoc tests with $p$-value adjustment using a multivariate $t$ distribution using the lsmeans() function in the R package emmeans.

Relative fitness of plasmid mutants was analysed in a linear model with Box-Cox transformation. Although the effect of the *lacZ* marker was found to be nonsignificant (1-sample $t$ test, $t_5$ = 0.32, $p$ = 0.76), fitness values were initially corrected for marker effects to give $W_{corr}$ by dividing each measurement by the mean fitness of plasmid-free strains (1.01). A linear model was performed with plasmid and marker and their interaction as fixed effects. Model reduction was performed using stepwise AIC comparisons and F tests. We detected no interaction effect of the marker ($F_{1,25}$ = 0.03, $p$ = 0.86). Post hoc pairwise contrasts were performed as above.

Relative fitness of plasmid mutants was analysed in a linear model, with plasmid and knock-out and their interactions as fixed effects, with post hoc contrasts performed as above.

Relative fitness of strains expressing PQBR57_0059 in trans, or the pQBR57 *parAB* genes in trans, were each analysed with LMMs, with transformant as a random effect, and pUCP variant and pQBR variant and their interactions as fixed effects. Model reduction was performed using stepwise AIC comparisons and likelihood ratio tests, and post hoc tests were performed as above. The experiments with pQBR103 and the pQBR57 *parAB* genes were analysed using linear models, as transconjugant replicates were not repeatedly measured.

The dynamics of different PQBR57_0059 variants over time were analysed using an LMM. We used Box-Cox–transformed cumulative plasmid density as the response variable and PQBR57_0059 status as a fixed effect. PQBR57_0059 variant was modelled using a random effect.

Growth curves were analysed using the ratios of cumulative densities of IPTG-induced and noninduced conditions as the response variable, and pME6032 insert as a fixed effect. Transformant replicate was modelled as a random effect. Post hoc contrasts were performed for each strain against the no-insert control using the trt.vs.ctrl option in lsmeans(), with $p$-value adjustment using the multivariate $t$ distribution.

Data and example analysis scripts are available at https://github.com/jpjh/COMPMUT/ and at the University of Liverpool Datacat, doi: 10.17638/datacat.liverpool.ac.uk/1275.

## Results

### Mutations to PFLU4242 and pQBR57_0059 emerged in parallel in evolution experiments

Previous evolution experiments with *P. fluorescens* SBW25 and megaplasmids pQBR103, pQBR57, or pQBR55 have identified 4 key genes implicated in plasmid compensatory evolution in this strain [29,33,38,40,53] (**Fig 1**; full details and discussion in **S1 Text**): the *gacA/S* 2-component signalling pathway, which has repeatedly been associated with pQBR103 and pQBR55 carriage; *PFLU4242*, an accessory chromosomal gene that has been mutated in

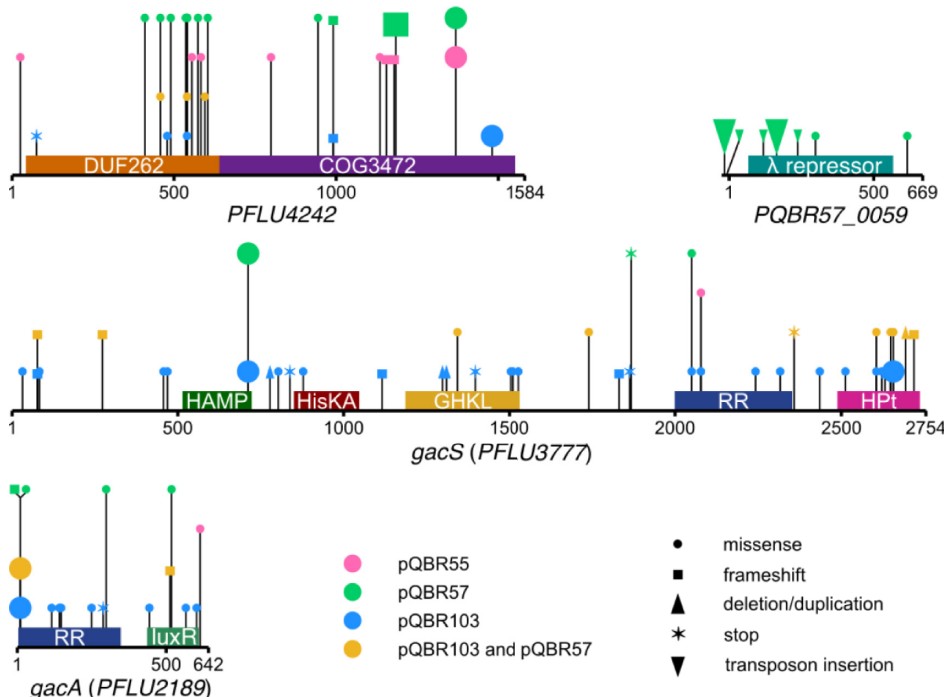

**Fig 1. Targets of compensatory mutation.** Diagrams show the 4 genes associated with parallel compensatory mutations from Harrison and colleagues [33], Hall and colleagues [29,40,54], and Carrilero and colleagues [38]. Symbols indicate the location of the mutation in the nucleic acid sequence. The colour of the symbol indicates the plasmid with which that mutation was associated, and the shape of the symbol indicates the type of mutation. Amino acid identical mutations (or identical insertion sites) that emerged in parallel in >1 population for that plasmid are shown as a larger symbol. Predicted domains are indicated as coloured bars above each sequence. COG, cluster of orthologous genes; DUF, domain of unknown function; luxR, luxR-type; RR, response regulator; other abbreviations follow PFAM convention (pfam.xfam.org). The full list of mutations across these studies, as well as a discussion of how these genes were identified, is provided in **S1 Text**. The data underlying this figure may be found at https://github.com/jpjh/COMPMUT/.

evolution experiments with all 3 pQBR plasmids tested; and *PQBR57_0059*, the only plasmid-borne gene that was a target of parallel evolution in pQBR57 [54]. Parallel mutations are a strong signal of selection [55,56], and the appearance of similar mutations in experiments with different plasmids led us to hypothesise that mutations to *PFLU4242* and *gacS* were general mechanisms of plasmid compensation in *P. fluorescens* SBW25. Given the difficulties of isolating wild-type pQBR55 transconjugants (i.e., without de novo compensation [29]), we focused on identifying the basis of fitness cost and amelioration in the 2 larger megaplasmids, pQBR57 and pQBR103. These mercury resistance plasmids were originally acquired by exogenous isolation from the same agricultural site in Oxfordshire, and, although distinct by RFLP analysis, genome sequencing revealed some distant gene similarity and synteny between pQBR103 and pQBR57 [47]. Sequenced pQBR plasmids do not clearly conform to established incompatibility types, but pQBR103 and pQBR57 have been shown experimentally to be compatible [38].

## Mutations to *PFLU4242*, to *gacS*, or to pQBR57_0059 ameliorate the costs of plasmid carriage

First, to test whether *PFLU4242* and *gacS* mutations indeed had a general effect on plasmid cost, we generated pQBR57 and pQBR103 transconjugants in either wild-type bacteria or in

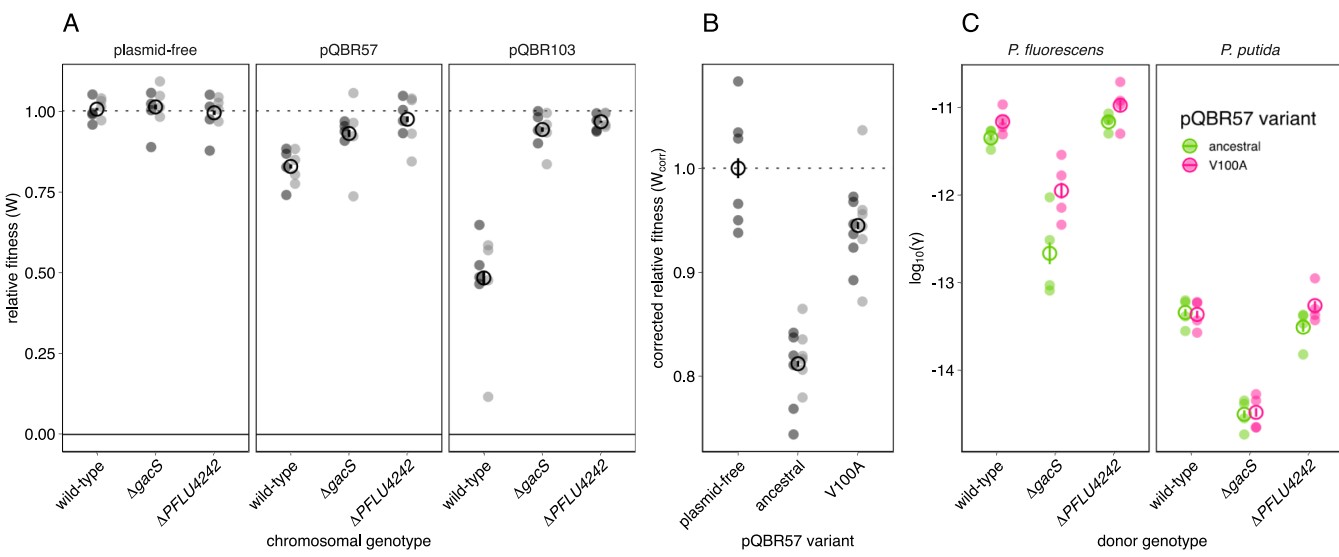

**Fig 2. Relative fitness and conjugation rates of plasmid bearing strains and putative compensatory mutations. (A)** Fitness of wild-type, Δ*gacS*, and Δ*PFLU4242* strains of *P. fluorescens* SBW25 carrying either no plasmid (left panel), pQBR57 (middle panel), or pQBR103 (right panel). All strains were competed against an isogenic, differentially marked plasmid-free wild-type *P. fluorescens* SBW25. Experiments were performed with 4 independent transconjugant strains as replicates, and were performed twice (light and dark grey). Each measurement is plotted, and the mean across all measurements for each condition indicated with an empty black circle, with error bars corresponding to standard error. **(B)** Fitness of strains with ancestral and V100A variant pQBR57 plasmids. Strains were competed against isogenic, differentially marked plasmid-free strains. Experiments were performed with a total of twelve independent transconjugants into either gentamicin-resistant (dark grey, *n* = 6) or streptomycin-resistant (light grey, *n* = 6) recipients. Relative fitness was corrected for marker effects as competitions were performed with both streptomycin- and gentamicin-resistant test strain backgrounds. For panels **A** and **B**, relative fitness of 1 (horizontal line) indicate no fitness difference between strains. **(C)** Conjugation rates of ancestral and V100A variant pQBR57 plasmids from wild-type, Δ*gacS*, or Δ*PFLU4242* strains of *P. fluorescens* SBW25 into either isogenic *P. fluorescens* or into *P. putida* KT2440. Experiments were performed with 4 independent transconjugant strains as replicates. Conjugation was measured according to Simonsen and colleagues [42]. For all panels, each measurement is plotted, and the mean across all measurements for each condition indicated with an empty circle, with error bars corresponding to standard error. The data underlying this figure may be found at https://github.com/jpjh/COMPMUT.

mutant backgrounds where we had deleted *PFLU4242* or *gacS* and measured fitness by direct competition against a plasmid-free isogenic strain (**Fig 2A**). Knockouts were used since the diversity of naturally arising mutations to each target (including deletions) was indicative of loss of function (**Fig 1**). Consistent with our hypothesis, we detected a significant effect of mutation on the fitness costs of plasmid carriage (LMM, plasmid:host interaction likelihood ratio test (LRT) $\chi^2$ = 37.3, $p <$ 1e-6). Each plasmid levied a significant cost in the wild-type background, corresponding with previous studies [38,47]: for pQBR57, 17.8% ($t$ = 4.8, $p$ = 0.003) and for pQBR103, 52% ($t$ = 10.2, $p <$ 0.001). Mutation of *PFLU4242* resulted in substantial amelioration such that we did not detect a significant fitness cost of either plasmid (post hoc contrasts with multivariate $t$ ("mvt") adjustment pQBR57 $t$ = 0.87, $p$ = 0.99; pQBR103 $t$ = 1.19, $p$ = 0.95), as shown previously [38]. Mutation to *gacS* was more equivocal, with more effective compensation of pQBR103 than of pQBR57 [38]: pQBR57-carrying Δ*gacS* mutants were neither significantly different from plasmid-free strains ($t$ = 2.1, $p$ = 0.5) nor from wild-type pQBR57-carriers ($t$ = 3.14, $p$ = 0.09). Neither the Δ*gacS* nor the Δ*PFLU4242* mutants had a detectable fitness difference from wild-type (*gacS* $t$ = −0.24, $p$ = 1; *PFLU4242* $t$ = 0.27, $p$ = 1), indicating that the effects of mutation were due to interaction with the plasmid rather than providing a direct fitness benefit.

To investigate the role of PQBR57_0059, we focused our attention on 2 sequenced, evolved pQBR57 variants from an evolution experiment [40]. One of these, A.01.65.G.002 ("anc") was identical to the ancestor, while the other B.09.65.G.029 ("V100A") had acquired only a single

basepair mutation across its 307,330 bp length, resulting in a Val100Ala mutation in the predicted carboxyl-terminal domain of PQBR57_0059. By conjugating the A.01.65.G.002 ("anc") and the B.09.65.G.029 ("V100A") evolved pQBR57 variants into the ancestral strain of *P. fluorescens* SBW25, and each of the chromosomal mutants, we could specifically measure the effects of this mutation (**Fig 2B**). Competitions showed that the V100A mutation largely ameliorated plasmid cost (ANOVA, effect of plasmid $F_{2,27} = 49.8$, $p = 8.65e-10$; post hoc contrasts with mvt adjustment anc versus V100A, $p < 1e-8$; plasmid-free versus V100A, $p = 0.037$), but that possessing this plasmid mutation provided no additional benefit in either the *gacS* or the *PFLU4242* mutant backgrounds (post hoc contrasts, effect of plasmid mutation given chromosomal mutation ($|t| < 1.7$, $p > 0.74$) (**S1 Fig**). We also tested 10 additional PQBR57_0059 variants that emerged during the previously reported evolution experiment [40,54] and likewise found that, in general, plasmids with a disrupted PQBR57_0059 allele were significantly better at persisting in the soil microcosm environment in which they had evolved (LMM, effect of PQBR57_0059 disruption $\chi^2 = 5.6$, $p = 0.02$, **S2 Fig**, **S1 Text**), although variance was high between evolved plasmids, probably due to second-site mutations in pQBR57.

Together, these data confirm that mutation to the chromosomal gene *PFLU4242* effectively ameliorates the cost of both pQBR57 and pQBR103 (as we previously showed both for these plasmids [38] and for pQBR55 [29]), and, similarly, that mutation to the plasmid gene *PQBR57_0059* ameliorates the cost of pQBR57. As megaplasmid cost can be ameliorated by disruption to just 1 gene (*PFLU4242*, *gacS*, or *PQBR57_0059*) our results suggest that the major burden of these megaplasmids comes not from the maintenance of plasmid DNA but rather from an interaction that depends on the functioning of specific genes.

## Plasmid compensatory evolution need not negatively affect plasmid transmission

Next, we tested whether the different modes of plasmid compensatory evolution affected conjugation rates. For this, we focused on pQBR57, since both chromosomal and plasmid-borne genes are implicated. We measured both intraspecific (between isogenic *P. fluorescens* SBW25 strains) and interspecific (to *Pseudomonas putida* KT2440) conjugation (**Fig 1C**). Consistent with previous measurements made in soil [19], conjugation into *P. putida* occurred much less readily than between *P. fluorescens* (linear model, effect of recipient $F_{1,43} = 925$, $p < 1e-8$). The *ΔgacS* mutation had a significant effect, reducing both intraspecific and interspecific conjugation efficiency by approximately 10-fold (effect of chromosomal mutation $F_{2,43} = 110$, $p < 1e-8$). However, neither the *ΔPFLU4242* nor the PQBR57_0059_V100A mutations significantly reduced conjugation, within or between species ($p > 0.14$ for both modes of compensation). This suggests that there is no necessary trade-off between vertical and horizontal modes of plasmid replication and that compensatory mutations can potentially enhance both maintenance and spread of conjugative plasmids.

Interestingly, our previous study [33] that identified GacA/S mutation as a route to pQBR103 amelioration did not find a significant reduction in conjugation rates in the evolved lineages. However, the lack of significance in that study was driven principally by replicates that had mutations in PFLU4242 rather than GacA/S, consistent with our findings with the clean knockout strains reported here.

## Distinct pQBR plasmids have common and divergent effects on chromosomal gene expression

The observation that mutations to *PFLU4242* or *gacA/S* ameliorate the costs of different pQBR plasmids suggests that there are convergent physiological responses to the carriage and

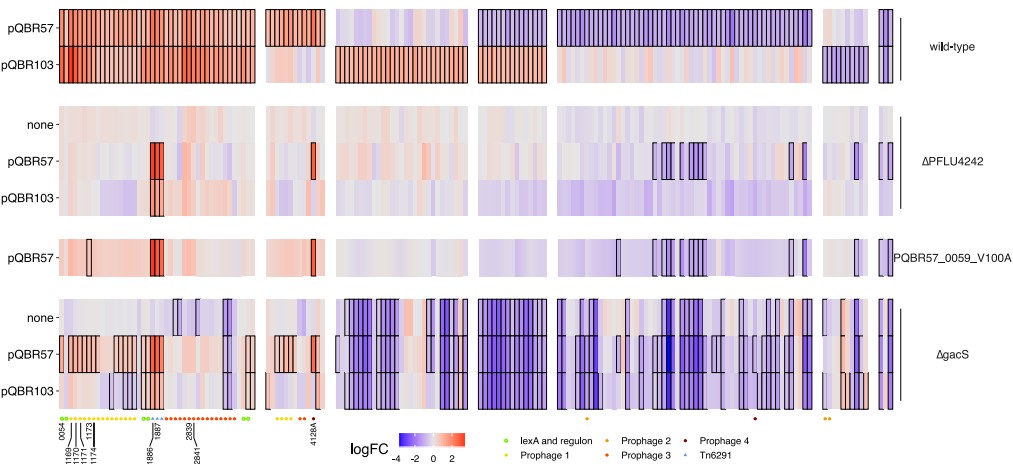

**Fig 3. Common and divergent effects on chromosomal gene expression following plasmid acquisition and amelioration.** Each column in the heatmap refers to a differentially expressed (FDR < 0.05, absolute $\log_2$ fold-change > 1) gene in bacteria carrying either plasmid relative to plasmid-free. Note: There were no additional genes meeting these criteria in the $\Delta PFLU4242$ or PQBR57_0059_V100A ameliorated strains; however, there were 401 additional differentially expressed genes as a result of *gacS* deletion, which have not been shown for clarity. Red indicates increase, and blue indicates decreases in expression; cells with a solid border had FDR < 0.05 for that specific condition. Columns have been grouped by pattern of response to pQBR57 and pQBR103. A coloured symbol below each gene indicates whether it is associated with a chromosomal MGE and/or the SOS response. Genes selected for cloning are labelled. FDR, false discovery rate. The data underlying this figure may be found at https://github.com/jpjh/COMPMUT/.

compensation of these different plasmids. To understand these responses and their resolution, we performed RNA sequencing (RNA-seq) on plasmid-free bacteria and from bacteria carrying either pQBR57 or pQBR103, either without amelioration mutations, or with the $\Delta PFLU4242$ or $\Delta gacS$ knockouts, or the PQBR57_0059 V100A mutation.

First, we analysed the effect of plasmid carriage without amelioration. Both plasmids caused extensive changes to gene expression, but the different plasmids had some common and some divergent effects on chromosomal gene expression (**Fig 3, S3 Fig**). pQBR57 affected the expression of 398 genes (FDR < 0.05), with 47 up-regulated >2× and 60 down-regulated >2×. pQBR103 affected expression of 254 genes, with 77 up-regulated >2× and only 11 down-regulated >2×. "Gene Ontology" (GO) terms enriched among the up-regulated genes for both plasmids included those associated with the SOS response and signal transduction. However, there were divergent effects of each plasmid, particularly for 33 genes that were up-regulated by pQBR103 but down-regulated by pQBR57. Carbohydrate metabolism was overrepresented in this set (7/33 GO:0005975 "carbohydrate metabolic process," $p_{adj}$ = 0.008), suggesting that while there was a common response to megaplasmid acquisition, each plasmid also had specific effects [26].

A closer examination of the differentially expressed genes shows a set of 50 genes that were at least 2-fold up-regulated by both plasmids. Many of these were co-localised in the genome, indicating co-regulation, and the possible activation of particular operons by pQBR103 and pQBR57. Of these 50 genes, 31 were in putative prophages, while 3 were in the Tn6291 transposon [40]. Since prophage induction is a signature of the SOS response, and the *P. fluorescens* SBW25 *lexA* homologues *PFLU1560* and *PFLU3605* were also up-regulated by both pQBR57 and pQBR103, we investigated how many of the 16 remaining up-regulated genes were located downstream of a predicted LexA binding site. By scanning the genome for the conserved pattern CTGKMTNNWHDHHCAG [57], we identified 12 genes <151 bp downstream of a predicted LexA site, of which 5 were up-regulated by both plasmids. The common transcriptional

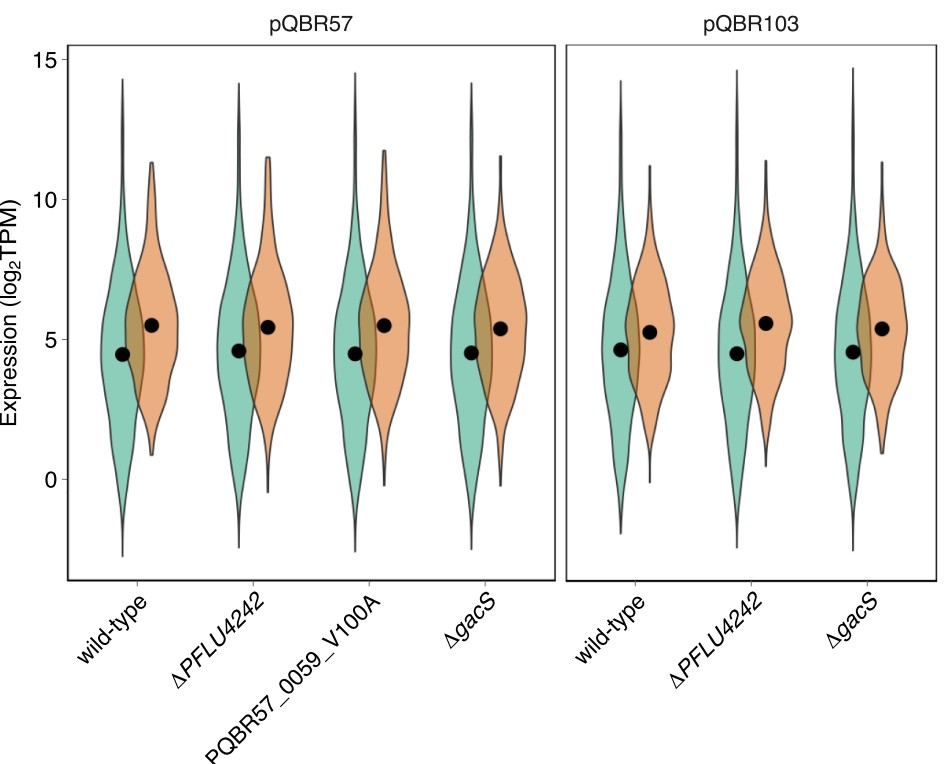

**Fig 4. Megaplasmid genes are expressed, and overall levels of expression remain the same regardless of amelioration.** Violin plots show log-transformed TPM of chromosomal (green) and plasmid (orange) genes, averaged across experimental replicates. Solid dots indicate the grand mean for that replicon and condition, calculated from log-transformed values. TPM, transcripts per million. The data underlying this figure may be found at https://github.com/jpjh/COMPMUT/.

response to pQBR57 and pQBR103 carriage by *P. fluorescens* SBW25 can thus be characterised as activation of the SOS response and chromosomal mobile genetic elements, which constitute 42/50 (84%) of the >2-fold up-regulated genes. This is not the same as the transient SOS response caused by plasmid acquisition reported by Baharoglu and colleagues [58], as the strains used in these experiments are many generations removed from the original transconjugant due to periods of growth during colony isolation and culture.

Plasmid genes were transcriptionally active. pQBR103 carries 478 predicted open reading frames (ORFs), while pQBR57 carries 426; for both plasmids, the majority of ORFs have unknown function. In the ancestral host, expression of pQBR103 ORFs ranged from 0.9 to 2,323 transcripts per million (mean 87, median 39), while expression from pQBR57 ranged from 1.8 to 2532 (mean 132, median 43). In general, plasmid-borne genes were expressed at levels approximately 2× higher than those of the chromosome, after accounting for gene length (**Fig 4**). In both cases, the most highly expressed genes from the plasmids were uncharacterised proteins (**S4 Fig**).

## Plasmid amelioration has broadly similar effects on transcription regardless of mechanism

We then investigated how compensatory mutations affected the changes in transcription caused by plasmid carriage.

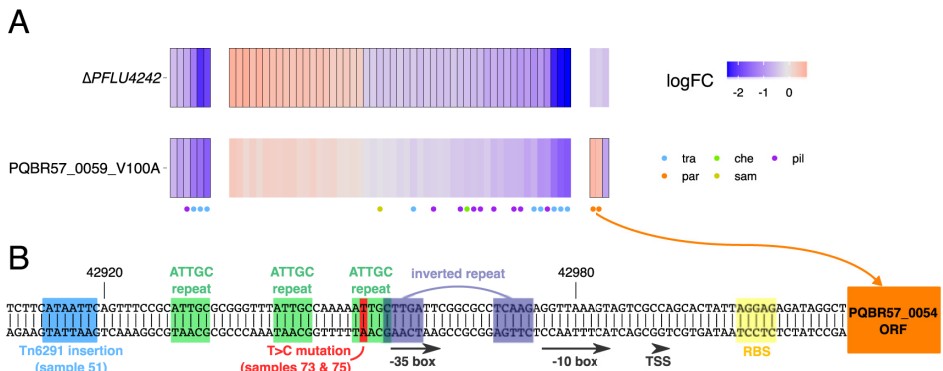

**Fig 5. Common and divergent effects on pQBR57 gene expression following different modes of compensatory evolution. (A)** Heatmap of differentially expressed (FDR < 0.05) genes on pQBR57 following either chromosomally located amelioration via ΔPFLU4242 or plasmid-located amelioration via the V100A mutation to PQBR57_0059. Each column in the heatmap refers to a differentially expressed (FDR < 0.05, absolute log₂ fold-change > 1) gene following amelioration. Red indicates increase and blue indicates decreases in expression; cells with a solid border had FDR < 0.05 for that specific condition. A coloured symbol below each gene indicates the predicted plasmid region affected: tra = transfer region (locus tags PQBR57_0201–0212); par = partition region (PQBR57_0054 and 0055); che = chemotaxis region (PQBR57_0044–0049); sam = SAM-associated region (PQBR57_0084–0099); pil = type IV pilus region (PQBR57_0074–0083) [47]. **(B)** The region upstream of the predicted pQBR57 par genes. Predicted transcription and translation factor binding sites are highlighted, as are identified repeats, and mutations identified by Hall and colleagues [40]. FDR, false discovery rate; RBS, ribosome binding site; TSS, transcription start site. The data underlying this figure may be found at https://github.com/jpjh/COMPMUT/.

## Effects of PFLU4242 on plasmid-mediated transcriptional responses

Without pQBR57 or pQBR103, disruption of *PFLU4242* had no detectable effect on chromosomal gene expression compared with wild-type bacteria ($p_{adj}$ > 0.05 for all genes). However, this same mutation in plasmid-bearing bacteria had a striking effect on plasmid-induced changes in gene expression: The reversion to ancestral levels of almost all (pQBR57: 376/398, 94%; pQBR103: 251/254, 99%) of the chromosomal genes differentially regulated following plasmid acquisition (**Fig 3**), including all of the SOS-associated genes up-regulated by both plasmids, and the majority of plasmid-specific differentially expressed chromosomal genes.

Plasmid genes were also differentially expressed following amelioration by PFLU4242 mutation, but only a subset of them (pQBR57: 57/426; pQBR103: 85/478). The responses of pQBR57 and pQBR103 were largely divergent, with PFLU4242 disruption generally resulting in down-regulation of pQBR57 genes, particularly those encoding putative conjugation apparatus (*tra*) and a type IV pilus (*pil*) (**Fig 5**), while pQBR103 genes were up-regulated by PFLU4242 disruption (including *tra* and *pil*) (**S7 Fig**). Although modulation of plasmid gene expression is likely to have some effect on cell physiology, these divergent responses lead us to conclude that the effects of plasmid transcriptional burden or plasmid gene expression *per se* are not primarily responsible for the costs of these plasmids, since the phenotypic costs of carriage could be abolished by *PFLU4242* disruption without major convergent effects on plasmid gene expression.

Notably, expression of the 3 genes from Tn6291 (*PFLU1886-1888*) was unaffected by PFLU4242 mutation amelioration for either plasmid. *PFLU1887* and *PFLU1888* are hypothesised to encode the transposase subunits for Tn6291, whereas *PFLU1886* is of unknown function. The fact that these genes remain up-regulated in the compensated strains suggests that these are not causally responsible for the main negative effects of plasmid carriage and also suggests that they are activated through a different route to the other plasmid up-regulated genes.

PFLU4242 mutation therefore had the general effect of bringing chromosomal gene expression back to levels similar to plasmid-free bacteria, suggesting that wild-type *PFLU4242* is instrumental in interacting with pQBR plasmids to generate a fitness cost.

## Effects of *gacS* on plasmid-mediated transcriptional responses

Compared with *PFLU4242*, unpicking the specific transcriptional effects of *gacS* deletion on plasmid carriage was more difficult, as *gacS* deletion by itself caused differential expression of 733 chromosomal genes affecting numerous phenotypes (see also [59]). One hypothesis was that disruption of GacA/S signalling ameliorates the cost of plasmid carriage by independently down-regulating genes that were up-regulated by the plasmids, thus "balancing out" the effects of plasmid acquisition. However, while we found a number of genes that were differentially expressed in the plasmid bearers and the *ΔgacS* mutant, the direction of expression varied between pQBR57 and pQBR103, and of the 50 genes up-regulated by both plasmids (logFC>0, FDR < 0.05), only 6 were independently down-regulated by the *ΔgacS* mutation, providing little support for this hypothesis.

We therefore considered an alternative hypothesis: that there exists an interaction between GacA/S signalling and the pQBR plasmids that is responsible for plasmid cost, and disruption of the GacA/S system prevents this interaction, resulting in amelioration. One pathway for an interaction between GacA/S and the plasmids could be through PFLU4242. The GacA/S system operates by interfering with the activity of inhibitory mRNA-binding Rsm proteins, allowing translation of that mRNA to occur. *PFLU4242* is predicted to have binding sites for RsmA, suggesting that *PFLU4242* may form part of the GacA/S regulon [60]. Indeed, we found that in the *ΔgacS* mutant, *PFLU4242* transcription was reduced by approximately 50%, but *gacS* expression was not affected in the *ΔPFLU4242* mutant (**S5 Fig**). These data provide an explanation for some of the convergence between these 2 mutational routes to compensation, namely that PFLU4242 is (more) proximally responsible for plasmid cost and that disruption to GacA/S could reduce plasmid cost by reducing expression of PFLU4242. Consistent with this hypothesis, and with the phenotypic fitness data, GacA/S disruption affected plasmid-induced changes in gene expression by reverting some of the differentially expressed genes back to ancestral levels, but this effect was incomplete, particularly with pQBR57 (**Fig 3**). To test directly whether increased expression of *PFLU4242* alone could recapitulate fitness costs in the *ΔgacS* mutants, *PFLU4242* was cloned into the inducible *Pseudomonas* expression vector pME6032 [61] and introduced into the *ΔgacS* mutant. The plasmids pQBR57 or pQBR103 were introduced, and the effects of *PFLU4242* expression on growth were tested by comparing growth curves between induced and uninduced cultures. Consistent with our model, induction of *PFLU4242* recapitulated a fitness cost in the *ΔgacS* mutant, but only in the presence of the pQBR plasmids (**S6 Fig**). The fact that PFLU4242 is expressed under wild-type plasmid-free conditions, along with experimental data showing no effect of *PFLU4242* overexpression in pQBR-plasmid-free strains (**S6 Fig**), also suggests that PFLU4242 by itself is not toxic, and it is interactions with the pQBR plasmids that underlies the fitness consequences of this gene.

Disruption of GacA/S signalling affected gene expression from both plasmids in a manner similar to PFLU4242 disruption, but there was an additional effect on pQBR57 that showed 26 genes, mainly of unknown function, specifically down-regulated in the *ΔgacS* mutant relative to the wild type (**S4 and S7 Figs**). There was no overall pattern in the predicted functions of these genes, other than their location near the putative origin of replication, and start of the sequence, of pQBR57.

The overlap between plasmid and gac-regulated genes likely represents a complex interaction between the plasmids and the gac system. However, the direct effect that GacA/S has on

*PFLU4242* transcription can effectively provide a unified explanation for the 2 chromosomal modes of amelioration, converging on PFLU4242.

## Effects of PQBR57_0059 on plasmid cost and amelioration

Mutations to PQBR57_0059 had a broadly similar effect on chromosomal gene expression to PFLU4242 (Fig 3). Each amelioration had similar effects on gene expression from pQBR57 (Fig 5). Compared with the plasmid-borne PQBR57_0059 mutation, the effects of the *ΔPFLU4242* chromosomal mutation were clearer, with many genes trending in a similar direction with the plasmid-borne mutation but not exceeding the FDR < 0.05 significance threshold. One clear exception were the *par* genes, which were up-regulated by the PQBR57_0059_V100A mutation but not by the *ΔPFLU4242* mutation (see below).

## A mechanism for plasmid cost and its amelioration

Together, our transcriptomic data indicate a set of genes that are up-regulated by both plasmids and then down-regulated by the different pathways of compensatory evolution. Expression of these genes therefore correlates with the phenotypic fitness costs of plasmid carriage. These genes may be responsible for physiological disruption and fitness costs, but it is also possible that their expression is a downstream effect of other disruptive interactions emerging as a consequence of plasmid acquisition. To test whether expression of these genes causally affects bacterial fitness, we selected 12 candidates for further investigation. Eight candidates were selected as genes that were >2× up-regulated by both plasmids in the wild type but not the *ΔPFLU4242* background, and 4 additional genes that did not meet these criteria were also selected as controls (Fig 3). The ORF of each gene was cloned into the inducible *Pseudomonas* expression vector pME6032 [61], and the effects of expression on growth were tested by comparing growth curves between induced and uninduced cultures (Fig 6).

We identified one gene, *PFLU1169*, which, when expressed, resulted in a considerable reduction in growth representative of a very large fitness cost (LMM effect of insert $\chi^2 = 49.2$, $p < 0.0001$; Dunnet post hoc test *PFLU1169* versus empty vector $t_{26} = 5.36$, $p < 0.0001$). Indeed, this gene was difficult to clone and express as growth in culture imposed strong selection for mutation of the insert or loss of the expression vector. *PFLU1169* is part of the *P. fluorescens* SBW25 "prophage 1" locus; however, further investigation revealed that this prophage is in fact a tailocin: a protein complex related to a phage tail that acts as a bacteriocin against competitors [62]. *PFLU1169* encodes the holin, expression of which, in concert with other

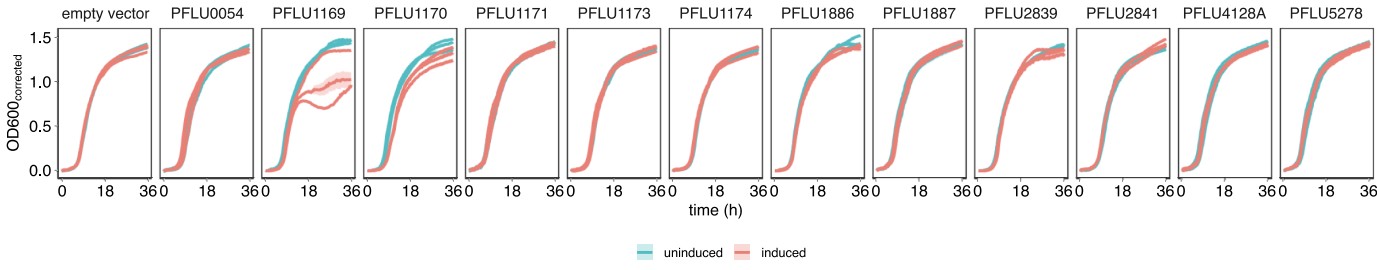

**Fig 6. Expression of genes up-regulated by pQBR plasmids imposes a fitness cost.** Different panels indicate cloned genes. Each vector was transformed into *P. fluorescens* 3 times independently, and the growth of each transformant was tested with and without IPTG induction 4 times, with the mean (solid line) and standard error (shaded area) calculated and plotted for each transformant separately, under 100 μg/ml IPTG (red) or uninduced (blue) conditions. One of the 3 pME6032::*PFLU1169* transformants did not show a large growth defect, but resequencing this plasmid from *P. fluorescens* revealed a 16-bp deletion in the insert in this replicate. No other mutations were identified in the other replicates, nor in any of the other inserts. The data underlying this figure may be found at https://github.com/jpjh/COMPMUT/.

prophage genes, causes cell lysis and tailocin release [63]. The prophage 1 locus is itself regulated through the SOS response, as a LexA binding site was found upstream of this operon. *PFLU1170*, another "prophage 1" gene, had a more marginal effect (*PFLU1170* versus empty vector $t_{26}$ = 3.2, $p < 0.03$). Overexpression of any of the remaining 10 genes had no detectable effect ($p > 0.99$ for all comparisons) under the tested conditions.

We have therefore identified one causal link between plasmid-induced gene expression changes and fitness costs. Plasmid carriage activates the SOS response, which activates the "prophage 1" tailocin, causing increased cell permeability and lysis. Lysis is likely to be stochastic, the result of which is an apparent reduction in fitness at the population level.

## pQBR57_0059 interacts with other plasmid genes to generate a fitness cost

PQBR57_0059 is predicted to encode a DNA-binding protein of the lambda repressor family (HHPRED top hit 3BDN "lambda repressor," $p$ = 4E-23, [64]), with the closest blastp matches in the NCBI "nr" database annotated as being either from other *Pseudomonas* species, or *Acinetobacter*. Lambda-like repressors are found in diverse bacteria and in other plasmids and prophage and operate by binding to repeat regions to prevent transcription [65]. We therefore considered it unlikely that PQBR57_0059 was a toxin that directly interfered with cellular physiology. Instead, we predicted that it affected the expression of other genes, either on pQBR57, on the chromosome, or both, resulting in a negative effect on competitive fitness. To distinguish between these alternatives, we cloned either the ancestral-type ("anc") or mutant ("V100A") variants of the PQBR57_0059 ORF under the control of the $P_{lac}$ promoter, which is constitutively active in *Pseudomonas*, on the expression vector pUCP18 [43] and introduced these variants to *P. fluorescens* SBW25. We then introduced either the evolved variant of pQBR57 with the V100A mutation or the unmutated variant. If the costs associated with wild-type PQBR57_0059 were due to direct interaction with the cell or solely through effects on chromosomal gene expression, fitness costs would be apparent by expressing the ancestral type variant from pUCP18. However, if other pQBR57 genes were involved, then pQBR57 would be required to recapitulate the effect, and the wild-type *PQBR57_0059* would complement the otherwise costless V100A pQBR57 variant.

Expressing just the wild-type PQBR57_0059 from pUCP18 was insufficient to generate a fitness cost without pQBR57 (Fig 7; LMM, post hoc tests for pUCP18 variants without pQBR57 $p > 0.91$ for all comparisons), and other genes on pQBR57 were required for PQBR57_0059 to be costly (interaction between pUCP18 and pQBR57 variant $\chi^2$ = 29.4, $p$ = 1e-4). Specifically, the presence of an otherwise costless V100A pQBR57 variant in cells expressing wild-type PQBR57_0059 in trans from pUCP18 restored the fitness cost (post hoc test $t_{22}$ = 4.97, $p$ = 0.0026, compare x-axis "no insert" and "ancestral" between "no pQBR57" and "V100A" panels). In fact, expression of PQBR57_0059 from pUCP18 appeared to decrease the fitness of ancestral-pQBR57-carriers even further (effect of anc versus no insert $p$ = 0.0078), probably due to the increased effect of higher PQBR57_0059 expression from the multicopy expression vector. Expressing the V100A mutant PQBR57_0059 in trans was not devoid of cost when ancestral pQBR57 was present ($p$ = 0.01), because the mutated protein likely has some residual activity that becomes apparent at high expression levels, but it had no significant effect in the presence of the V100A variant pQBR57 ($p$ = 0.18). Together, these data show that the fitness effects of PQBR57_0059 are likely to emerge from its effects on other plasmid genes, hinting that a more complex interaction between the plasmid and chromosome is responsible for the cost of pQBR57.

We therefore investigated plasmid-borne genes that were specifically differentially expressed by the PQBR57_0059 amelioration. Among the plasmid genes differentially

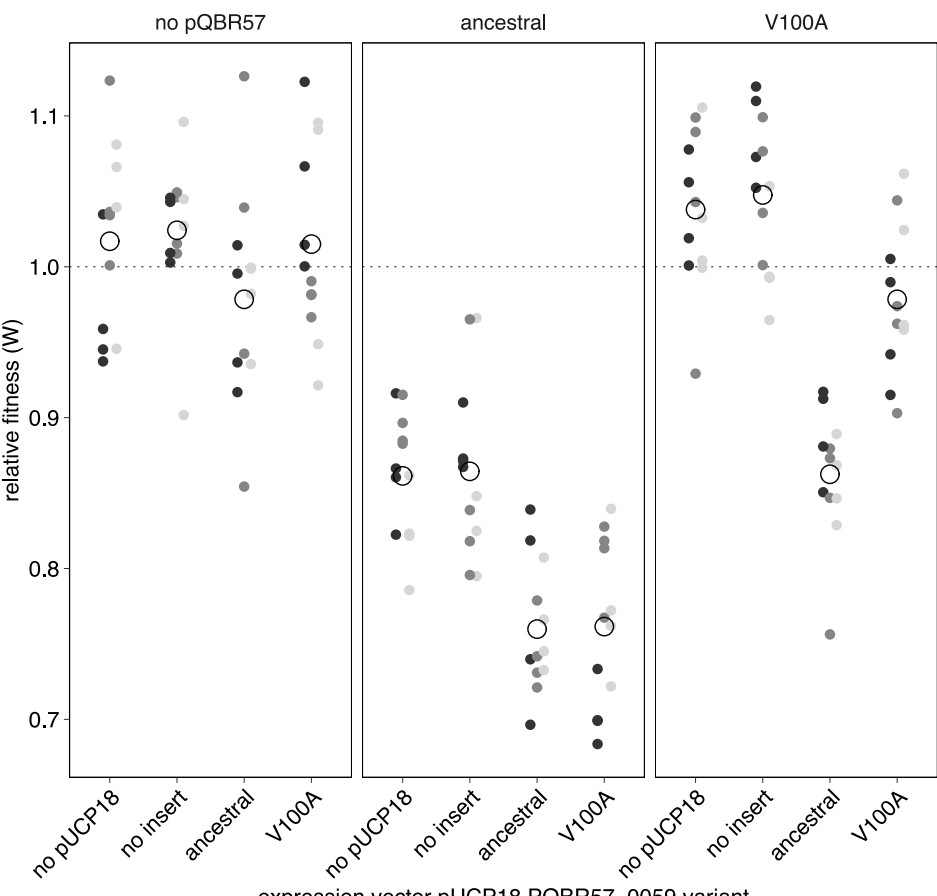

**Fig 7. PQBR57_0059 interacts with other pQBR57 genes to generate a fitness cost.** Relative fitness as determined by direct competition for *P. fluorescens* SBW25 overexpressing either the ancestral or the V100A variant of PQBR57_0059 (or no-insert expression vector, or no-vector controls, and carrying either no pQBR57 or the ancestral or V100A variants of pQBR57, against a plasmid-free wild-type *P. fluorescens* SBW25 without expression vector. Competitions were performed with 3 independent transformants/transconjugants indicated by shades of grey, and the fitness of each transconjugant/transformant was tested 4 times. For the "none/no-pQBR57" control treatment, all measurements were taken on the same strain. The data underlying this figure may be found at https://github.com/jpjh/COMPMUT/.

expressed by amelioration, only 3 were specifically affected by pQBR57_0059 (**Fig 5A**). One, PQBR57_0215, was down-regulated in a similar direction by the *ΔPFLU4242* mutation ($0.05 < p_{adj} < 0.1$) and thus is likely to be a general response to amelioration. However, the other 2, which form an operon PQBR57_0054–0055, were exclusively up-regulated by PQBR57_0059 disruption. These genes are predicted to encode a ParAB system (HHPRED top hit for PQBR57_0055 6SDK "Spo0J/ParB" $p$ = 2E-24; PQBR57_0054 matches 2OZE_A "ParA/Walker-type ATPase" $p$ = 4.2E-25), homologues of which have been found to be responsible for efficient segregation of plasmids to daughter cells [66]. Two other genes resembling ParB are found in pQBR57 (PQBR57_0051 and PQBR57_0316), but neither has an adjacent gene matching ParA. Although we do not know whether the PQBR57_0054–0055 operon constitutes a functional segregation system in pQBR57, these matches and the adjacency of these genes, combined with their similarity and synteny with the ParAB genes of pQBR103 (locus tags pQBR0001 and pQBR0002), led us to label PQBR57_0054–0055 a putative *par* system for pQBR57 [47]. Examination of the region upstream of this operon reveals 3 ATTGC

repeats and an inverted repeat (CTTGA(N)$_9$TCAAG) that might interact with pQBR57_0059, which is predicted to bind DNA (**Fig 5B**). Interestingly, this region was also the target of parallel mutations in a previous evolution experiment [40]: In one population, evolved pQBR57 carried by *P. putida* had acquired an insertion of Tn6291 transposon 78 bases upstream from the predicted transcriptional start site, while in another replicate, evolved plasmids carried both by *P. fluorescens* and *P. putida* gained a single T>C transition 38 bases upstream of the transcriptional start site. Together, this suggests that pQBR57_0059 acts as a repressor of the *par* locus, and disruption of this repressor, or disruption of its binding site, results in increased transcription. The fact that there is no additive benefit of mutations in PFLU4242 and pQBR57_0059 suggests that increased par gene expression somehow prevents a deleterious interaction with PFLU4242, directly or indirectly, to ameliorate plasmid cost.

## Increased expression of the *par* genes from pQBR57 is sufficient to ameliorate the costs of pQBR57, but not pQBR103

To test the hypothesis that increased pQBR57 *parAB* gene expression as a consequence of the disruption of the repressor PQBR57_0059 was responsible for ameliorating the costs of pQBR57, we cloned the putative *par* genes PQBR57_0054–0055 into pUCP18 and introduced this vector, or an empty pUCP18 control, into *P. fluorescens* SBW25. We then introduced pQBR57 and measured the effects on relative fitness. Consistent with our hypothesis, overexpression of the *par* genes ameliorated the costs of pQBR57 (**Fig 8A**; LMM, interaction of pUCP18 insert and pQBR57 carriage $\chi^2$ = 22.2, $p < 1e\text{-}4$; pairwise contrasts no insert pQBR57 versus *par* pQBR57 $t_{10}$ = 5.15, $p = 0.004$) but had no significant fitness effects by themselves (**Fig 8A**, no insert no pQBR57 versus *par* no pQBR57 $t_{10}$ = 1.82, $p = 0.5$). Although *par* genes are normally associated with distribution of plasmids to daughter cells, we found no significant effect of *par* overexpression on the emergence of plasmid-free segregants in serially transferred populations, indicating that the beneficial effects of *par* expression on the fitness costs of pQBR57 are not directly mediated by plasmid loss (**S8 Fig**; $p > 0.05$ for all time points and plasmids).

We then investigated whether overexpression of pQBR57's *par* genes had a general effect on plasmid fitness costs. We introduced pQBR103 to the strains that were overexpressing PQBR57_0054–0055 and again measured relative fitness. In contrast to the amelioration of pQBR57, expression of PQBR57_0054–0055 had no beneficial effect on pQBR103 (**Fig 8B**, LM, pUCP18:pQBR plasmid interaction $F_{2,24}$ = 1.79, $p = 0.18$; pairwise contrast pQBR103 no insert versus pQBR103 *par* $t_{26}$ = 2.41, $p = 0.12$), and we had weak evidence that costs were in fact exacerbated (effect of pUCP18 insert $F_{2,26}$ = 4.4, $p = 0.02$). Increased expression of pQBR57 *par* therefore reduced the fitness costs of pQBR57, but did not have any ameliorative effect on a heterologous plasmid.

## Discussion

Plasmids are known to impose fitness costs, but the molecular causes of such costs are far from clear. Here, we show that, rather than emerging from the general bioenergetic demands of replicating DNA, transcribing into mRNA, or translating into protein [20], the majority of the fitness costs of plasmid acquisition come from specific deleterious interactions between plasmid and chromosomal genes. As a consequence, small mutations—in some cases just single base changes—can be sufficient to ameliorate even a very large plasmid with hundreds of genes. Previous studies have tended to focus on smaller, nonconjugative replicons [25,30,67], or have identified plasmid compensatory mutations with substantial pleiotropic effects: disruption or deletion of conjugative machinery from the plasmid (rendering it immobile) [28,68]; deletion

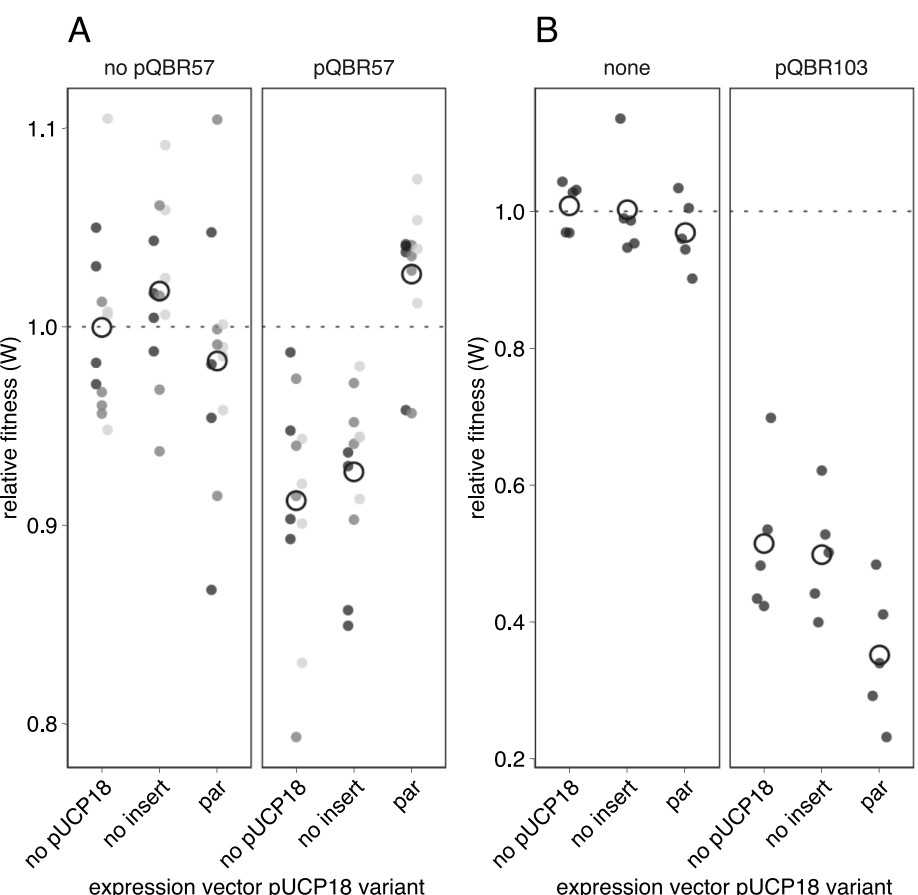

**Fig 8. Effects of pQBR57 par gene expression on megaplasmid cost. (A)** Competitions were performed with 3 independent transformants/transconjugants indicated by shades of grey, and the fitness of each transconjugant/transformant was tested 4 times. Empty circles indicate the grand mean for that set of conditions. **(B)** Expression of pQBR57's par genes in trans exacerbates the fitness costs of the distantly related megaplasmid pQBR103. Competitions were performed with 5 independent transconjugants, with each one tested once. In all cases, competitions were against a plasmid-free wild-type *P. fluorescens* SBW25 without expression vector. The data underlying this figure may be found at https://github.com/jpjh/COMPMUT/.

of large portions of the plasmid [34,69]; or disruption of extensive and multifunctional regulatory circuits [31,33,70]. In contrast, our data show that single mutations can enable large, natural, conjugative, plasmids to become rapidly accommodated with few obvious trade-offs, and furthermore begins to describe the molecular negotiation by which this occurs (**Fig 9**).

Specifically, we have linked the fitness costs of both pQBR57 and pQBR103 to the SOS response in *P. fluorescens* SBW25. SOS response–associated genes comprised 42/50 of the genes up-regulated in common, and we found that increasing expression of just one gene in the SOS regulon—the "prophage 1" holin *PFLU1169*—had a severe effect on fitness, implicating the SOS response as a direct proximal cause of plasmid-induced fitness costs. The SOS response has wide-ranging impact on bacterial physiology [71], meaning that *PFLU1169* is unlikely to be the sole mechanism of plasmid-induced fitness costs, perhaps explaining why this gene has not been detected as a mutational target in our previous evolution experiments. How is the SOS response triggered by the pQBR megaplasmids? Single-stranded DNA, such as is acquired in the early stages of conjugative transfer, can directly activate the SOS response

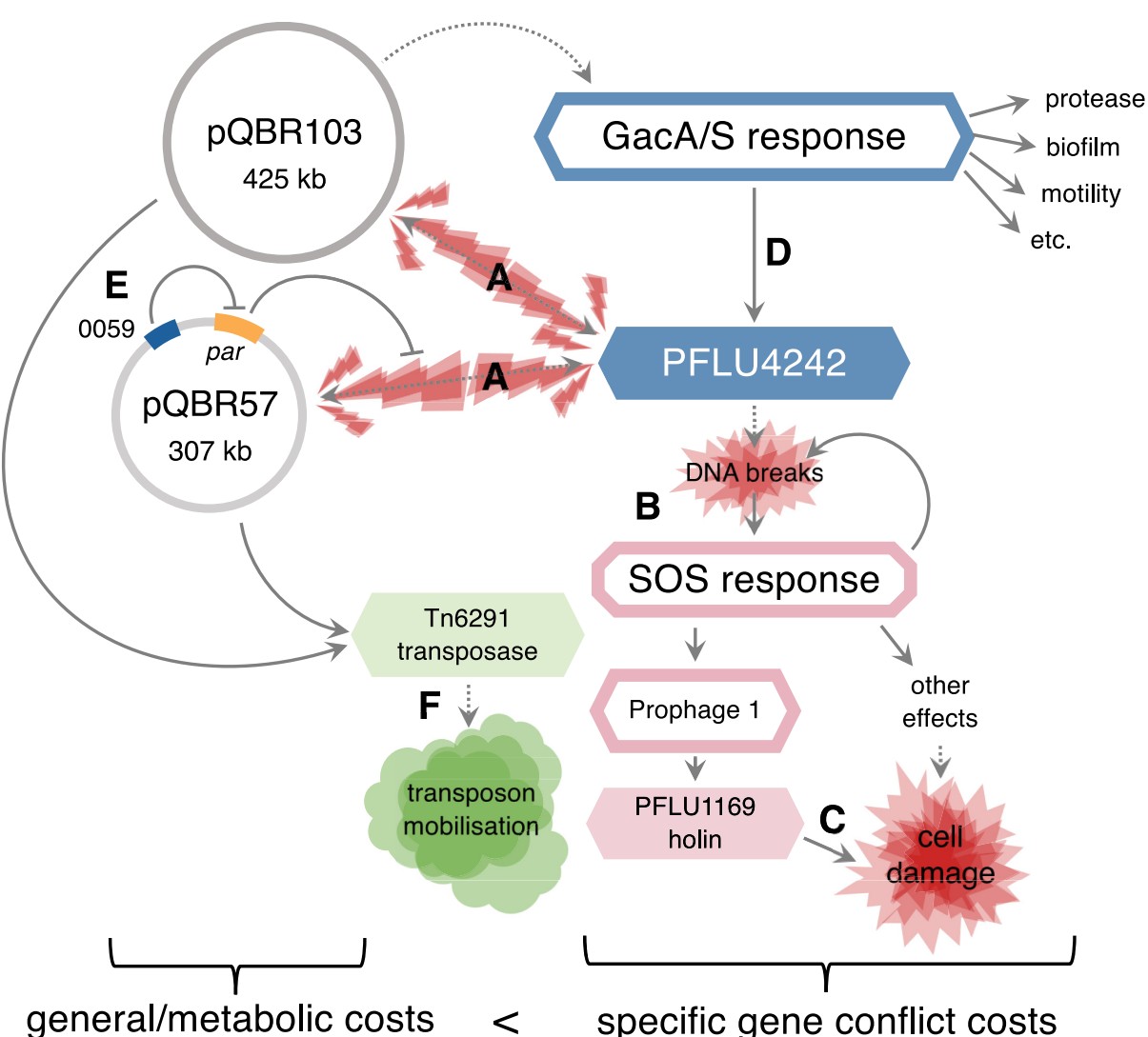

**Fig 9. Proposed interactions leading to pQBR plasmid fitness costs.** Dotted lines indicate speculative interactions, and solid lines indicate experimentally or bioinformatically determined interactions (from this study or others). Targets of compensatory mutation are shown in blue. Genes are solid boxes, whereas pathways/groups of genes are outlines. **(A)** Both plasmids interact with PFLU4242. **(B)** PFLU4242 activates the SOS response, probably through causing DNA breaks. **(C)** One outcome of the SOS response is expression of PFLU1169 that directly causes cell damage. **(D)** The effects of plasmid interactions with PFLU4242 are exacerbated by GacA/S, which increases expression of PFLU4242. pQBR103 may also interact directly with the GacA/S system. **(E)** Mutation of PQBR57_0059 increases expression of par, which reduces the disruption of pQBR57 but not pQBR103. **(F)** Both plasmids activate the Tn6291 transpose through a different route to PFLU4242 or GacA/S, which is likely to increase transposition activity and mobilisation of Tn6291. Overall, specific gene conflict costs greatly exceed the metabolic costs of pQBR57 and pQBR103.

[58], but as the strains in this study were many generations removed from the original conjugation event due to picking and restreaking of transconjugant colonies, the SOS response in the pQBR/SBW25 system is clearly being perennially activated through a different route. DNA damage is the most likely candidate, implicating the SOS response as essentially an amplifier of disruptions caused by plasmid acquisition. Like an amplification circuit, the SOS response can itself cause DNA fragmentation, forming a positive feedback loop, and further disruption [72].

How does plasmid carriage cause DNA breaks? Clashes between the machineries of transcription and replication put stress on DNA molecules, and such clashes have been shown to

drive the fitness costs of the minimal (2.1 kb) plasmid pIND in *E. coli* [67]. In *Shewanella onei-densis*, the replication machinery of the IncP1-beta mini-replicon pMS0506 (13.1 kb) probably triggers the SOS response by binding to the host DNA helicase DnaB and blocking the replication fork, exposing vulnerable ssDNA intermediates to breaks [73]. In *Pseudomonas aeruginosa* PAO1, the small (5.1 kb) plasmid pNUK73 triggered the SOS response due to increased expression of the plasmid replication machinery, which may cause breaks due to increased stress on DNA molecules within the cell [25]. However, other costly plasmids did not have the same effect on *P. aeruginosa* PAO1, suggesting that SOS response activation was not a general source of plasmid fitness costs in that strain [26]. The fact that different species of *Pseudomonas* had differing SOS responses to the same pCAR1 metabolic plasmid [74] likewise suggests that activation of the SOS response usually occurs in a plasmid-by-host specific manner, again consistent with our hypothesis that plasmid fitness costs emerge principally through specific genetic conflicts.

It is unclear whether the conflicts between transcription and replication seen with small plasmids is responsible for DNA damage for much larger plasmids like pQBR103 and pQBR57. We saw no significant change in megaplasmid copy number [40] or *rep* gene expression when comparing ancestral with ameliorated strains, hinting that other processes may be at work. What is clear is that mutations to a limited number of targets can prevent activation of the SOS response, presumably by preventing DNA breaks from occurring. Principal among these mutations, in terms of degree and scope of amelioration, are loss-of-function mutations to PFLU4242. While the molecular and evolutionary functions of PFLU4242 remain unknown, some clues can be gained from bioinformatic analyses. PFLU4242 contains 2 predicted domains: an N-terminal domain of unknown function 262 (DUF262) domain (HHPRED hit PF03235.16 $p$ = 2E-18), and a carboxyl-terminal DUF1524/COG3472 domain (HHPRED hit PF07510.13 $p$ = 4.6E-9) (**Fig 1**). This domain architecture resembles that of known endonucleases that target modified phage DNA [75], and related genes are located in known genome defence islands [76]. The hypothesis that PFLU4242 has a nuclease function in phage defence would explain how PFLU4242 disruption might prevent DNA breaks, and also why this gene has been maintained despite having no detectable effect on fitness or gene expression in plasmid-free bacteria in an axenic laboratory setting. Work is ongoing to understand the molecular and evolutionary functions of this gene. The concept of a trade-off between openness to HGT and resistance to phage infection has been explored extensively in the context of CRISPR/Cas and restriction modification modules [14,77,78]. Our data hint at another mechanism by which genome defence might have side effects: an inappropriately excessive response driving fitness costs, analogous to immune hypersensitivity.

We show that PFLU4242 expression is induced by a global regulator of secondary metabolism, the GacA/S 2-component system, and so our data are consistent with the hypothesis that disruption to GacA/S could exert its effect through repression of PFLU4242. It is possible that global regulators have been implicated in plasmid persistence due to their regulatory effects on more proximal causes of plasmid conflict [31,70]. The combined size of the *gacA* and *gacS* genes (approximately 3.4 kb) is over twice that of *PFLU4242* (approximately 1.6 kb) (**Fig 1**), providing a larger mutational target, which, alongside the mutability of the GacA/S system [79], may explain why GacA/S rather than PFLU4242 was the primary target of mutation in one of our previous studies [33]. On the other hand, loss of GacA/S function causes broad-scale phenotypic side effects beyond reduced fitness costs of carrying pQBR plasmids, such as reduced biofilm production, motility, and exoprotease secretion, each of which could impact bacterial fitness in more complex environments [59]. Notably, loss of GacA/S function was not detected in our previous experiments conducted in soil microcosms [54]—in such habitats, compensatory mutations targeting the genes causing host–plasmid conflicts appear to be

favoured over mutations affecting their multifunctional regulators. We also have hints that PFLU4242 is not the only mechanism by which pQBR plasmids and the GacA/S system interact. Carriage of pQBR103 has a similar effect on PFLU4242 expression as deletion of *gacS*, while 26 pQBR57 genes appear to be differentially regulated in the Δ*gacS* background but not in the Δ*PFLU4242* background. Moreover, pQBR103 appears to carry a homologue of the Gac-Rsm system signal transducer RsmA, in common with pQBR55 but not pQBR57 [47]. Understanding how plasmids may manipulate core pathways in bacterial signalling is an important subject for future investigation [80].

Compensatory mutations occurring on a plasmid are thought to have increased potential to enhance plasmid persistence and accelerate HGT relative to mutations of similar effect on the chromosome, because plasmid-borne compensation is inherited both by daughter cells and by transconjugants [36]. Mutation of the putative lambda-family repressor PQBR57_0059 caused increased expression of the putative partitioning machinery of pQBR57, reducing the cost of that plasmid. The molecular mechanism by which *par* expression reduced pQBR57's cost remains unclear. Although they are best known as machinery for DNA segregation, partitioning proteins are known to be multifunctional: Chromosomal ParB is known to bind DNA both specifically (at *parS* sites) and nonspecifically, impeding access by DNA-damaging agents like nucleases, forming bridges and loops between distant parts of a DNA sequence, and affecting gene expression [81–86]. The putative ParB of pQBR57 may also act to prevent DNA damage, perhaps by restricting nuclease access to otherwise vulnerable DNA. What, then, is the evolutionary function of PQBR57_0059 in repressing the protective *par* module? Tuning *par* expression for plasmid cost reduction may be a delicate matter, as our data suggest that high levels of pQBR57 *par* expression alone has a cost (as has also been found for chromosomal ParB expression [82,85]) and may exacerbate the costs of heterologous plasmids as we see for pQBR103. A highly conjugative plasmid such as pQBR57 is expected to frequently find itself in co-existence with other plasmids [87], and PQBR57_0059 may be one mechanism by which pQBR57 can modulate its own activity in multireplicon cells. Other plasmids are known to carry genes that prevent cellular disruption, for example, the anti-SOS genes PsiB [88] or the nucleoid-like proteins H-NS that repress plasmid gene expression [89], reducing costs. In this context, it is interesting that we have witnessed the evolution of this "stealth" function for the pQBR57 *par* genes during the course of our evolution experiment, showing that plasmids can contain the genetic resources necessary to evolve their own amelioration.

The principal group of genes up-regulated by both plasmids were located in chromosomal regions annotated as mobile genetic elements. Foremost among these were prophages 1 and 3. Plasmid activation of putative prophage genes appears to be a relatively common response to plasmid acquisition, having also been observed following acquisition of the IncP-7 plasmid pCAR1 by *P. putida* KT2440, *P. aeruginosa* PAO1, and *P. fluorescens* Pf0-1 [27,74]. However, the prophages of SBW25 are not predicted to be intact integrated temperate phage and rather seem to be relics that have lost the genes involved in DNA replication and packaging [63,90,91]. These prophages are therefore candidate tailocins—phage tails that have been evolutionarily co-opted to act as toxins used in inter-strain competition [62]. To exert their activity, tailocins translated in the cytoplasm must be released by cell lysis [62]. Tailocin production is therefore an example of bacterial spite, whereby an individual suffers a fitness cost to harm a competitor [92]. Spite, like altruism, is evolutionarily stable only within a narrow range of conditions—specifically, where those benefitting from the action are likely to be related and those targeted by the action are unlikely to be related [93]. Receiving a new plasmid is a clear signal of the presence of an unrelated competitor and thus an opportune time for spiteful tailocin activation. The SOS response can be considered a strategy against aggressions, rather than a solely defensive reaction [71]. It is therefore tempting to speculate, that, rather than being

solely a side effect of DNA damage, expression of tailocins following plasmid acquisition could be an adaptive trait, one which both inhibits competitors [93] and potentially also prevents invasion of parasitic plasmids through cell suicide in a manner similar to abortive infection [94]. Interestingly, the prophage up-regulated by pCAR1 in *P. fluorescens* Pf0-1 appears also to be a tailocin [63], suggesting that toxin induction in response to plasmid acquisition is not specific to this bacterial–plasmid pairing.

The other chromosomal mobile genetic element up-regulated by both pQBR57 and pQBR103 was the transposon Tn6291. The up-regulated genes are thought to encode the transposase, which provide the machinery enabling Tn6291 to relocate both to other chromosomal locations and, importantly, to conjugative plasmids. Tn6291 does not appear to encode conjugative machinery of its own and therefore depends on other apparatus to move horizontally between bacterial lineages. Indeed, we previously showed using experimental evolution that Tn6291 exploits pQBR57 to move between *Pseudomonas* species in soil [40]. This dependence of Tn6291 on other mobile genetic elements for transmission provides an adaptive explanation for why the transposase genes are sensitive to conjugative plasmid acquisition: The presence of a conjugative plasmid signals the presence of a vehicle for Tn6291 to use. Communication between the pQBR plasmids and Tn6291 appears to be independent of the SOS response, as the Tn6291 transposase remains up-regulated in the otherwise ameliorated plasmid-bearing strains. There are numerous adaptive reasons why mobile genetic elements might be expected to communicate with one another, since conflict and collaboration between mobile genetic elements is expected to be a key factor influencing HGT and hence bacterial evolution (e.g., [2,95]). Interpreting this language could prove key to understanding the factors driving the spread of ecologically and clinically important traits.

More broadly, our data are consistent with the growing body of evidence that implicates interactions between horizontally acquired genes resident in the chromosome and incoming plasmids as a key source of conflicts potentially leading to fitness costs. While we identified a positive interaction between Tn6291 and the pQBR plasmids, other studies have found conflicts between plasmids and genomic islands. A series of studies have described how acquisition of plasmid pCAR1 by *P. putida* KT2440 induced expression of ParI, a ParA-like gene located on an accessory chromosomal genomic island, which, in that case, interfered with the plasmid-encoded ParAB system to destabilise pCAR1 [27,96,97]. MGE-encoded *par* homologues can clearly have contrasting effects on plasmid success. Genomic island-encoded helicases represent another group of horizontally acquired DNA-binding proteins that have been implicated in conflicts in diverse bacteria-plasmid associations involving *Pseudomonas* [25,32]. While *PFLU4242* does not appear to be a helicase, it is clearly part of the accessory genome, as homologues are not found in related strains of *P. fluorescens*. In fact, conflict between plasmids and bacteria may be more accurately described as conflict between plasmids and resident MGEs, with bacterial cells merely the battleground.

Our work has focused on large plasmids, generally exceeding 150 kb in size, sometimes called megaplasmids. Megaplasmids are likely to be especially effective vectors of HGT as they can mobilise many traits at once—a particular concern for the spread of antimicrobial resistance because selection with one drug may drive the spread of multidrug resistance [98]. Similarly, compensation of megaplasmids potentially facilitates the maintenance of many different traits in a lineage, provided that (consistent with our findings here) costs do not emerge from plasmid size per se. Advances in long-read sequencing technologies are beginning to expose the diversity, ubiquity, and complexity of megaplasmids in diverse bacterial genera, and, therefore, our findings are likely to apply beyond this specific case of mercury resistance in a soil Pseudomonad (e.g., [99,100]). For example, pQBR103 and pQBR57 are distant relatives of a large family of clinically important megaplasmids that appear to be disseminating multidrug

resistance in clinical isolates of *P. aeruginosa* from opportunistic infections [101,102]. In those strains, these megaplasmids appear to be stable even without the selective pressure of antimicrobials, suggesting that those strains are either preadapted for megaplasmid carriage or have undergone prior compensatory evolution.

The fact that plasmid costs emerge principally from specific gene conflicts is not to say that replication and expression of plasmid genes per se levies no burden nor that such costs are invisible to selection. Lynch and Marinov [20] calculated the costs of bacterial gene replication, transcription, and translation and found that the costs of even replicating short pieces of non-beneficial DNA are sufficient for purifying selection in moderately sized bacterial populations, with the costs of transcribing and translating those genes several orders of magnitude greater. The assays we used to measure fitness have limits, and it is likely that the residual bioenergetic burden of plasmid carriage would be detected by more sensitive methods [103]. Nevertheless, it is clear that such bioenergetic costs are many fold smaller than those that emerge from conflicts between specific genes or their products and may well be overwhelmed by other selective pressures in natural environments (such as phage [104]) or rendered negligible in the context of HGT [105].

Our findings have important implications for understanding the evolution of bacterial genomes. Fitness costs of plasmid carriage are, in theory, a barrier to HGT; however, our data suggest that when such costs are caused by specific genetic conflicts, they will be readily and rapidly ameliorated by single compensatory mutations thus enabling the long-term maintenance of plasmids in bacterial genomes. This helps to explain why plasmids are so common in bacterial genomes and, moreover, suggests that such compensated plasmid-carrying lineages will become important hubs of HGT within bacterial communities [19,106]. The fact that plasmid costs depend on specific conflicts is relevant to understanding the dissemination of ecologically and clinically important bacterial traits, such as antimicrobial resistance, because the immediate fitness costs of the encoding plasmids are unlikely to predict their long-term dynamics: Purifying selection against plasmids will diminish over time due to rapid compensatory evolution. More generally, the dynamics of mobile elements in bacterial genomes are unlikely to be predictable from their generic properties and thus more sophisticated analyses of genetic associations in pangenomes (e.g., [107]) will be required to predict the patterns and outcomes of gene exchange and how these will shape pangenome dynamics.

## Supporting information

**S1 Fig. There is no additional benefit of carrying both plasmid and chromosome compensatory mutations.** Relative fitness as determined by direct competition for ancestral and V100A variants of pQBR57, in wild-type, Δ*gacS*, or Δ*PFLU4242* strains of *P. fluorescens* SBW25, against plasmid-free wild-type *P. fluorescens* SBW25. Competitions were performed with 4 independent transconjugants. Each measurement is plotted and the mean across all measurements for each condition indicated with an empty black circle, with bars indicating standard error. The data underlying this figure may be found at https://github.com/jpjh/COMPMUT/.
(EPS)

**S2 Fig. Naturally occurring PQBR57_0059 mutations improve pQBR57 persistence.** Serial passage of ancestral strains bearing mutated plasmids from the evolution experiment, in competition against plasmid-free recipients in soil microcosms. Panels show the relative proportions of different subpopulations over 5 transfers for 3 independent replicates (rows a, b, and c). Columns correspond to different pQBR57 variants that were conjugated into an ancestral strain for this experiment. Columns are arranged according to whether that variant had a

disruption in, or immediately upstream of, PQBR57_0059 (right-hand columns) or not (left-hand columns). All detected mutations in that variant are described at the top of the column, with numbers corresponding to the affected locus_tag (with leading zeroes removed for reasons of space). "Tn" indicates a transposon insertion. Mutations in PQBR57_0059 are indicated in **bold**, and mutations affecting the region immediately upstream of PQBR57_0059 are in italics. Columns 1 and 2 contain data reproduced from Hall and colleagues [40]. Filled regions correspond to the proportions of recipients, transconjugants, donors, and segregants; note that very few segregants were detected in any population. The data underlying this figure may be found at https://github.com/jpjh/COMPMUT/.
(EPS)

**S3 Fig. Volcano plots showing changes to gene expression associated with plasmid carriage.** Points are coloured according to **Fig 3**. *PFLU1169* is highlighted with a black border. The data underlying this figure may be found at https://github.com/jpjh/COMPMUT/.
(EPS)

**S4 Fig. Gene expression varies across the plasmids, with the most highly expressed genes outwith regions with predicted functions.** Heatmaps show level of expression in TPM across each plasmid under different amelioration conditions. Coloured bars across the top indicate regions annotated in Hall and colleagues [47]: grey = rep; green = che; orange = par; purple = pil; yellow = SAM; red = Tn5042 mercury resistance transposon; pink = uvr; blue = tra. Loci are numbered according to the locus_tag feature. TPM, transcripts per million. The data underlying this figure may be found at https://github.com/jpjh/COMPMUT/.
(EPS)

**S5 Fig. Expression of *PFLU3777* (*gacS*) and *PFLU4242* in the different treatments.** Note different y-axis scaling. Points indicate different replicates, with values for each replicate scaled by effective library size. The data underlying this figure may be found at https://github.com/jpjh/COMPMUT/.
(EPS)

**S6 Fig. Overexpression of *PFLU4242* recapitulates pQBR plasmid fitness cost in Δ*gacS* mutants.** Columns of panels indicate whether the expression vector was empty or contained PFLU4242. Each vector was transformed into *P. fluorescens* ΔgacS 4 times independently before conjugation with plasmid pQBR57 or pQBR103. Independent replicates are indicated with thin lines, and the mean across replicates is indicated by the thick line, for each induction condition. Comparison of the effect of induction on areas under the curve showed that PFLU4242 expression had a significant effect in the presence of pQBR57 ($t_{5.04} = 27$, $p < 0.001$) or pQBR103 ($t_{3.33} = 5$, $p = 0.017$) but not for pQBR-plasmid-free ($t_{3.95} = -0.05$, $p = 0.96$). The data underlying this figure may be found at https://github.com/jpjh/COMPMUT/.
(EPS)

**S7 Fig. Differentially expressed plasmid genes across all amelioration conditions.** Image as **Fig 5A**, but showing the effect of Δ*gacS* and differentially expressed genes on pQBR103. The data underlying this figure may be found at https://github.com/jpjh/COMPMUT/.
(EPS)

**S8 Fig. Overexpression of pQBR57 par genes does not significantly affect plasmid loss.** Single colonies derived from independent transformations/conjugations were established in KB media supplemented with 50 μg/ml kanamycin (to select for maintenance of the pUCP18 expression vector), and transferred 1:100 into fresh media every 2 days for 4 transfers (approximately 25 generations). At each transfer, a sample was spread on KB agar, and after colonies

had grown, was replica-plated onto KB supplemented with 100 μM HgCl$_2$. Plots show proportion of mercury-resistant colonies at each time point. There was no effect of pQBR57 par overexpression on the emergence of segregants ($p > 0.05$ for all time points). The data underlying this figure may be found at https://github.com/jpjh/COMPMUT/.
(EPS)

**S1 Text. Supporting information Results and Discussion.**
(PDF)

## Acknowledgments

We would like to thank staff at the NERC Biomolecular Analysis Facility, University of Liverpool, for assistance with the RNA-seq and Jenna Gallie (Max Planck Institute for Evolutionary Biology) for advising on approaches to identify putative RsmA targets in *P. fluorescens* SBW25.

## Author Contributions

**Conceptualization:** James P. J. Hall, Ellie Harrison, A. Jamie Wood, Steve Paterson, Michael A. Brockhurst.

**Data curation:** James P. J. Hall.

**Formal analysis:** James P. J. Hall.

**Funding acquisition:** James P. J. Hall, Ellie Harrison, A. Jamie Wood, Steve Paterson, Michael A. Brockhurst.

**Investigation:** James P. J. Hall, Rosanna C. T. Wright.

**Methodology:** James P. J. Hall, Michael A. Brockhurst.

**Project administration:** Michael A. Brockhurst.

**Resources:** Rosanna C. T. Wright, Katie J. Muddiman.

**Supervision:** Steve Paterson.

**Visualization:** James P. J. Hall.

**Writing – original draft:** James P. J. Hall, Michael A. Brockhurst.

**Writing – review & editing:** James P. J. Hall, Rosanna C. T. Wright, Ellie Harrison, A. Jamie Wood, Steve Paterson, Michael A. Brockhurst.

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
