## [Editor Report · Decision Letter 0]

23 Apr 2021

Dear Dr Hall, 

Thank you for submitting your manuscript entitled "Plasmid fitness costs are caused by specific genetic conflicts" for consideration as a Research Article by PLOS Biology.

Your manuscript has now been evaluated by the PLOS Biology editorial staff, as well as by an academic editor with relevant expertise, and I'm writing to let you know that we would like to send your submission out for external peer review. Please accept my apologies for the delay incurred while we sought external advice.

Please re-submit your manuscript within two working days, i.e. by Apr 27 2021 11:59PM.

Kind regards,

Roli Roberts

Senior Editor

PLOS Biology

---

## [Decision Letter · Decision Letter 1]

11 Jun 2021

Dear Dr Hall,

Thank you very much for submitting your manuscript "Plasmid fitness costs are caused by specific genetic conflicts" for consideration as a Research Article at PLOS Biology. Your manuscript has been evaluated by the PLOS Biology editors, an Academic Editor with relevant expertise, and by four independent reviewers.

You’ll see that everyone except perhaps reviewer #4, who questions the novelty, is very positive about your study. Overall, most of the requests are textual and/or presentational, with a handful of experimental requests. In case it helps guide your revisions, the Academic Editor said "Clearly, some of the additional experiments suggested by rev#2 (e.g. using GacA/S KO and checking for plasmid loss and transfer during competitions and filamentation using microscopy) might solidify conclusions, but these seem beyond the scope of this study. The point about novelty raised by rev#4 seems valid and the authors should at least respond to the suggestions to remedy this by the same rev."

In light of the reviews (below), we are pleased to offer you the opportunity to address the [comments/remaining points] from the reviewers in a revised version that we anticipate should not take you very long. We will then assess your revised manuscript and your response to the reviewers' comments and we may consult the reviewers again.

We expect to receive your revised manuscript within 1 month.

**IMPORTANT - SUBMITTING YOUR REVISION**

*Resubmission Checklist*

*Published Peer Review*

*PLOS Data Policy*

*Blot and Gel Data Policy*

Sincerely,

Roli Roberts

Roland Roberts

Senior Editor

PLOS Biology

rroberts@plos.org

REVIEWERS' COMMENTS:

Reviewer #1:

In this manuscript, JPJ Hall and colleagues investigate the origin of plasmids fitness costs using two different megaplasmids in Pseudomonas fluorescens. After a meticulous and extensive series of experiments the authors show that plasmids costs originate form the activation of the SOS response, which activates different MGE in the bacterial chromosome, including the expression of a chromosomal tailocin toxin. The authors were also able to dissect the connection between the different compensatory mutations (which alleviate the cost of this plasmid) and the actual source of costs. This manuscript is the culmination of several previous papers from this same group looking at compensatory evolution of plasmid-associated costs in the same experimental model system. In general, I think this is an excellent work, carefully designed, executed and interpreted. My only "relevant" criticism to this manuscript is that it may result a bit too long/dense, and while a plasmid biologists like myself has REALLY enjoyed reading it, a more "general" reader may have a little trouble navigating the paper. The authors could maybe try to streamline the manuscript (results and discussion) and maybe move some parts to the supplementary material (for example "Plasmid compensatory evolution need not negatively affect plasmid transmission"?).

Minor comments

Line 39, helping to explain?

Line 265 "of plasmid compensation"?

Line 374, close parenthesis?

Line 486, Figure 5 instead of Figure 4, which also affects the numbering of the following figures.

Figure 4A, full names in legend may help the reader.

The interaction between pQBR57_0059 and the upstream region of the parAb genes seems likely, but it would be great to confirm if with an electrophoretic mobility shift assay. I understand however that this may be beyond of the scope of this work. 

If tailocin PFLU1169 is a crucial source of plasmid costs one may expect mutations inactivating this particular gene in compensated clones. Have the authors ever found mutations there?

Figure 8 is important, because it summarises the numerous findings of the study, and of previous studies, in a single model, but some of the connections are a bit difficult to understand for me. For example: "(B) PFLU4242 activates the SOS response, probably through causing DNA breaks". The authors showed that inactivation of PFLU4242 produced the silencing of the plasmid-induced SOS response, but that does not necessarily indicate that PFLU4242 activates the SOS response. In fact the expression level of PFLU4242 does not increase in the presence of the plasmids (Fig S5). Its true that the authors speculate about the potential role of PFLU4242 an anti-phage nuclease, but I think that what has been shown for sure is that plasmids activate the SOS response, and that compensatory mutations silence it back. Therefore I think the arrows form the plasmids should maybe point directly the SOS response instead of to the GacAS and PFLU4242 (and these should be linked with the silencing of the SOS response), but this is just a suggestion.

Reviewer #2:

Jamie Hall et al. wrote up a clear manuscript about a very thorough and well laid out study that aimed at identifying the molecular mechanism of plasmid cost amelioration observed in their Pseudomonas strains that were evolved previously with various plasmids. Specific mutations had previously been observed, and here potential mechanistic explanations are provided. Since only a few studies have been able to really pinpoint the molecular mechanisms by which plasmid fitness cost can be improved over time, this is an important study. The authors did point to a tailocin that was upregulated due to a 'chronic' SOS response cased by the plasmid, explaining the plasmid's fitness cost. However the nuclease function of PFLU4242 is still speculative, yet intriguing as an anti-phage defense system. Similarly, the role of the par genes on the plasmid is intriguing but needs further study. Additionally, the paper contains several noteworthy findings: examples include different pathways of plasmid-host compensatory evolution converging to restore host transcription levels, and direct upregulation of a transposon by a plasmid (independent of SOS response).I have few major comments but a few suggestions or questions to improve clarity.

Major comments:

1. It is not very satisfying that higher expression of par genes on a plasmid would decrease the fitness cost of that plasmid, and not rather decrease the loss rate (see also comment 2: were the strains plated on medium with plasmid, or was the loss rate really unchanged?). The authors write in Fig. 7" Mutation of PQBR57_0059 increases expression of par, which reduces the disruption of pQBR57". In the discussion the authors suggest possible functions of these genes other than plasmid partitioning. Are the authors sure that the competition assays were not confounded by plasmid loss during the assays? And can they say a bit more about how similar the genes are to known partitioning genes and what the likelihood is that they are involved in partitioning, or rather may have a different function as speculated? 

2. L. 156 and further: What does it mean to measure the fitness of the plasmids? Of course competition experiments that allow to calculate the Malthusian fitness of a strain relative to another are not to measure the fitness of the 'plasmids'? Do you mean fitness cost on the host? The traditional method measures the fitness of a plasmid-containing host relative to the plasmid-free host. The verbiage as it stands requires more context and explanation. From lines 159 - 163 we are to understand plasmid transfer can account for up to 10% of the estimated plasmid population, and measurements of 'plasmid fitness' (?) can increase fitness by 7%...shouldn't we be quite concerned about the effect of these numbers on competition results? Also related: Figure 2 (B) has Y-axis of 'corrected' relative fitness… What does this mean? the way I understand it, you are tracking the two competitors based on their chromosomal mutations and you are ignoring that the plasmid may have transferred to the competitor or may have been lost? Loss seem negligible (but was it tested for both plasmids and also when complementing cloned genes?). A high (10%) transfer frequency would slow down the originally plasmid-free competitor with a costly plasmid. Thus in the case of competition between plasmid-bearing (P+) and plasmid-free (P-) strains, the cost of the plasmid would be underestimated (overestimating the relative fitness of P+)

3. Line 553: "...the direct effect that GacA/S has on PFLU4242 transcription can effectively provide a unified explanation for the two chromosomal modes of amelioration, converging on PFLU4242." This could be tested directly and would strengthen a paper seeking molecular mechanisms. Use the ΔGacS strain(pQBR) and express PFLU4242 from a vector: if fitness decreases, then it really is the downregulation of PFLU4242 in ΔGacS strains that ameliorated cost, eliminating the confounding effect of the 700+ genes under GacS control.

4. L. 810: "The principal group of genes upregulated by both plasmids were located in chromosomal regions annotated as mobile genetic elements." The authors should point out that potential conflict between a genomic island and a plasmid was (first?) pointed out by San Millan et al in P. aeruginosa and then Loftie-Eaton et al in Pseudomonas sp. The finding of potential conflict between chromosomally located and independently replicating mobile genetic elements in multiple species of Pseudomonas is intriguing.

5. L. 604 and possibly other places in the manuscript: "Plasmid acquisition activates the SOS response,...". This may be misinterpreted as the event of acquiring the plasmid activating the SOS response, but it is not. As the authors explain elsewhere: the plasmid activates the SOS response long after it was acquired horizontally by conjugation, a sort of 'chronic' SOS response. It may be better to call this "plasmid carriage".

6. It would be nice to show microscopically that individual cells with plasmid look different before amelioration - often cells are elongated due to SOS response, and cell lysis could be demonstrated with a live/dead staining.

Minor comments:

1. L 26: plasmids don't have 'expression levels'. Maybe 'plasmid gene expression levels', or 'copy number'.

2. Line 28: 'This single chromosomal locus acts as a key mediator of plasmid fitness cost'… for pQBR-like plasmids.L. 60: how do you distinguish between bacterial growth and replication? Can you clarify?

3. L. 265; mechanisms OF plasmid compensation.

4. Paragraph starting on L. 395: wasn't this already done by Harrison et al for one of these plasmids? More context would be helpful.

5. L 399: 'GO terms' is not defined? 

6. Fig. 4, L. 490: "red indicates..." comes right after you explain the "coloured symbols" below each gene, yet now red/blue refer to differential expression levels, this could confuse the reader; swap sentences?

7. L. 521: "causing translation": do the authors mean "allowing translation to occur"? Basically Rsm is not or less active, and thus does not inhibit translation (or less so)? And translation of what?

8. L. 559: difficult sentence; try to rephrase.

9. Fig. 6 and other relative fitness figures: it is not always clear against which strain these strains were competed against. The legend says "Relative fitness' but it doesn't say relative to what. In the Methods (L 150) the authors clearly state "For all competitions, GmR strains were competed against SmR-lacZ [43] strains.", but it may be helpful to remind the reader of that in the figure legends. Is it always the plasmid-free SmR-lacZ or sometimes the plasmid-containing?

10. I am a bit lost with Fig. 6: Was the strain with mutated plasmid (and with no insert or vector) indeed more fit than the same strain without plasmid? And the cloned mutated plasmid gene does not completely restore the fitness cost. This paragraph describing Fig. 6 needs to be more clear.

Reviewer #3:

In this work Hall et al. study the amelioration of megaplasmids resulting from coevolution with Pseudomonas. Expanding on multiple previous works the authors delve into the specific causes and mechanisms of plasmid cost and amelioration. They show that both plasmids induce the host's SOS response, which consequently promotes the expression of prophage genes that negatively affect bacterial fitness. Disruption of a hypothetical endonuclease, or its downregulation via the disruption of the gacA/S system, could prevent SOS induction and restore host fitness. Additionally, upregulation of the partition system of one of the plasmids was also able to alleviate that plasmid's cost. It was a pleasure to review this paper, the experimental work is well designed, the analyses are sound, the manuscript is written in a clear way and easy to follow, and the message is also novel. The work is very comprehensive, and I request no further experiments, but I would like to discuss a few points that may be relevant.

gacS deletion: 

From Fig 2A and S1B it is not clear to me if gacS deletion fully compensates the cost of pQBR57, could you specify the statistical results? 

In Fig 1C (for intraspecific conjugation) there seems to be a small effect from the interaction between the gacS deletion and PQBR57_0059 variants. Do the two PQBR57_0059 variants differ significantly in the gacS mutant?

PFLU4242 and par locus:

The statement in line 772 is only valid for pQBR57, correct? How do you reconcile the PFLU4242 mediated fitness cost in the pQBR103 carrying strain, given that this plasmid already downregulates PFLU4242 to the same level as the gacS deletion (lines 535 and 781)? 

Could this be the reason why the cloned par locus has no effect on pQBR103 amelioration? Interestingly, the par genes from pQBR103 also seem to be upregulated (less than 2-fold?) in the gacS mutant (fig S4 & S6). Would the native par system of pQBR103, instead of that from pQBR57, be able to ameliorate the cost of pQBR103 (line 692)?

Conjugation:

PFLU4242 disruption leads to upregulation of pQBR103 tra and pil genes. Could this explain why the conjugation rate of this plasmid did not increase (line 382)?

Tn6291:

Do the plasmids originally carry copies of Tn6291, or could they have acquired it during the experiment? Could transposition and consequently Tn6291 increased copy number explain the upregulation of Tn6291 genes (line 497)?

Discussion:

The statement in line 705 could be supported by the work of Hideaki Nojiri with pCAR1, specifically PMID: 17675379, where it is shown that chromosomal transcriptional changes are due to specific cross talk between plasmid and chromosomal encoded par genes. In that work, the one cited in reference 71 and PMID: 24889869, it is also shown that pCAR1 upregulates the expression of prophage genes. I think a discussion of these papers would enrich this manuscript.

The statement in line 707, "furthermore describes the molecular negotiation by which this occurs", may be too bold as this is not a full description since there are still some unknowns in the mechanism provided in Fig. 8.

The authors show in this work that plasmid costs are due to specific genetic conflicts, in this case SOS induction that consequently induces prophage genes. Would it be relevant to discuss the interplay between plasmids and accessory elements (single genes or mobile genetic elements such as phages) as a source of cost, rather than the plasmid-host interaction? Indeed, the fitness costs of other plasmids have been shown to derive from conflicts with horizontally transferred helicase genes (references 25 and 32).

In lines 23, 88-92, 363 and 774, the authors suggest that costs arising from general plasmid properties, such as the requirement of the replication and transcription/translation machineries, would require multiple or large (deletions) amelioration events, while specific gene interactions could be resolved by more targeted solutions. I hope I did not misunderstand the authors point of view. Assuming I understood it correctly, I only agree partially, due to the global effects of two component systems and other general transcription regulators. In the present work, for example, targeted solutions for the specific conflict exist such as the disruption of PFLU4242 and PQBR57_0059, but so does the disruption of gacAS. The latter, by regulating a variety of genes, could compensate simultaneously more general costs. Indeed, the initial interpretation of the results from the work in reference 33, could be that the proximal cause of plasmid cost was translational demand. The current manuscript however shows that the conflict is more specific. The point being that disruption of global regulators provides a confounding factor in what concerns the identification of the proximal causes of fitness costs, since both specific or general conflicts can be targeted. Therefore, this kind of interpretation is not trivial and the discussion of this matter should be more careful.

Lines 160 - 163: could you elaborate on this? I do not understand the message the authors try to convey with this information.

Methods:

Was conjugation performed in solid or liquid media (line 132)? I also did not see a methods section concerning the conjugation reported in line 367.

Figures and supplementary information:

In Fig S2, 12 pQBR57 variants are shown, 5 with an intact PQBR57_0059, and 7 with variants of this gene, but 10 variants are mentioned in line 334. Moreover, could clarify in the legend of Fig S2 what are the loci affected in variants A01, A11, C01, C10 and C11? The locus_tags alone are not very information, are all these encoding hypothetical proteins?

PFLU5278 (shown in Fig. 5) is not labeled in Fig 3.

There are two Figure 4 (lines 448 and 486) and correct the following references to figures: Fig. 2B (line 329), Fig. 5A. (line 644), Fig. 3 (line 419).

The reference in line 1239 is missing from the reference list in the supplementary information. Please refer to Fig. S2 in the section starting in line 1266.

Minor comments:

"reverse genetics" should simply be "genetics" (lines 27 and 115). Please be more precise genetic notation: "cloned as a fragment into the EcoRI/KpnI site of pUCP18" (lines 139 and 145), "SmR lacZ+ strains" (line 150), and "par locus" or "parAB genes" (lines 235, 239 and 665).

Line 55: Entry exclusion systems are features of other genetic elements, not bacterial host features. 

Vogwill and MacLean's (PMID: 25861386) meta-analysis suggests that plasmid size is not correlated with fitness cost. This could be discussed in line 75.

The statement in line 102 could be supported by reference 28 (Porse et al. 2016), while the conclusion in line 402 could be supported by reference 26 (San Millan et al. 2018).

It would be helpful to state plasmid relatedness early in line 256, as done in line 853.

Line 604 can be misinterpreted as plasmid acquisition immediately after conjugation. Replacing acquisition by carriage may avoid confusion.

A few typos: "orbital" (line 171), "acquisition" (line 395), "expression" (line 513), close the parenthesis (line 374).

Reviewer #4:

The study 'Plasmid fitness costs are caused by specific genetic conflicts' by Hall et al. describes mechanisms of plasmid fitness cost reduction in Pseudomonas species. The authors describe four major locations of mutations that appear during the coevolution of host and plasmids. The authors further analyze the effect of these mutations on host fitness and find that one particular mutation elevates the cost of two different plasmids. The authors also analyze the effect of plasmid carriage on the transcriptome on the cell, which they combine with the effect of the observed mutations on gene expression. Analyzing the transcriptome, the authors find that most effects are overall plasmid specific, and mainly effect the SOS response of the cell. They further find that one of the plasmid compensatory mutations leads to downregulation of most chromosomal SOS genes. 

The study represents an interesting example of host-plasmid interaction in a natural model system. Furthermore, the authors reveal through transcriptomics the importance of stress-regulated response-receptors for the persistence of plasmids. Nonetheless, the study does not feel well connected in many parts yet and the expectation of the title is unfortunately not fully met in the results. 

Before going scientific comments on the study I would like to raise one point that made me concerned about the results in the study: 

The authors claim in the abstract and introduction (and discussion line 808) that they conduct 'experimental evolution', yet I couldn't find any evolution experiments (only competition over time?) but rather only citations of previous studies. Using previous results is not wrong and even a good practice in my opinion, but should definitely not be hidden from the reader that is given a false impression of the study. 

Furthermore, reading the previous study by Hall et al (DOI: 10.1099/mic.0.000862), reveals the discovery of the mutations in PFLU4242 and its biological description (fitness impact). Thus, taking both points together I remain concerned about the novelty of parts of the here presented results. 

The authors should consider restructuring their manuscript to have a larger focus on the indeed very striking (and very exciting) transcriptomic analysis. One suggestion could be to start with transcriptomics (as the mutations are already published material) and have more emphasise on the mutated genes (maybe by phylogenetic analysis) and the ectopic cloning of the by transcriptomics identified candidate genes. 

Major comments: 

1) For the beginning of the manuscript, the authors should consider a deeper analysis of the genes in which compensatory mutations appear. For example, for PFLU4242, which is indeed an intriguing candidate. Here, I am missing a deeper description of the PFLU4242 gene itself. What predicted domains does it have (hhpred?)? I see some annotation in figure one, but am missing explanation in the text (or better representation of the domains). 

In addition, does this gene have homologues in other species and. In general, it is not fully clear whether PFLU4242 is a housekeeping gene or rather may have accessory function. A deeper analysis in pseudomonas could help to understand the role of the gene. 

The same relates to the plasmid gene PQBR57_0059? Does is have homologs in other plasmids or phages? The authors should aim to draw a larger picture of the meaning of the genes and mutations here. 

2) In previous publications (e.g. Harrison et al.,2015) mutations in gacA/S had a strong negative impact on the host cell. How do the other interpret that they don't observe that in the current study? Such result would fit the observed changes in the transcriptome. In addition, the authors could speculate that mutations gacS/A may be not sustainable and other mutations (like in PFLU4242) might be more 'feasible' for the cells. The authors should include this in their discussion. 

3) The persistence of the different plasmid mutations (Fig.S2). Many points are unclear in this figure. 1)Where are these mutations coming from? And what does 'naturally emerging' (line 334) mean? 2) Maybe I missed this information, but are these additive mutations? 3) The coloring is not clear, are there segregants (black) anywhere in the figure? The authors need to clarify the results of this figure in the text and the figure itself. 

4) There is some mistake in the numbering of the figures (2x figure 4). The Figure 4(2) is not clear and needs to be modified. 

5) Maybe I overlooked it, but in Fig. 5, the authors write corrected OD600. I couldn't find anything in the methods section. What did the authors correct for? This needs to be clarified.

6) Regarding figure 8 - How does the growth curve look of strains carrying the large plasmids that activate the tailocin? Is it comparable in regards to toxicity/ fitness decrease? In addition, should lysis not be visible in form of plaques on solid media? The authors should clarify. 

Minor points:

1)How similar are the presented plasmids? It is important to state that these are (I assume) different rep types and in general plasmids of different origin. 

2) Figure 1 is very small (and not very intuitive), especially the writing. 

3) Did the authors ever attempt to study the effect of double (or consecutive) mutations? It might be not in the framework of this study, but perhaps the authors have data for it. 

4)There is some mistake in the numbering of the figures (2x figure 4)

---

## [Decision Letter · Decision Letter 2]

3 Sep 2021

Dear Dr Hall,

Thank you for submitting your revised Research Article entitled "Plasmid fitness costs are caused by specific genetic conflicts" for publication in PLOS Biology. I have now obtained advice from three of the original reviewers and have discussed their comments with the Academic Editor. 

Based on the reviews, we will probably accept this manuscript for publication, provided you satisfactorily address the remaining points raised by the reviewers. Please also make sure to address the following data and other policy-related requests.

IMPORTANT:

a) Please address the remaining requests from reviewer #3.

b) Please provide a more informative title; maybe "Plasmid fitness costs are caused by specific genetic conflicts that can be easily compensated" or Plasmid fitness costs are caused by specific genetic conflicts rather than intrinsic plasmid properties"

c) Many thanks for providing the data and code needed to reconstruct the Figures in Github. Please could you cite this location for the data clearly in each relevant main and supplementary Figure legend? e.g. "The data underlying this Figure may be found in https://github.com/jpjh/COMPMUT/" (we suggest using this rather than the institutional repository which is likely to be less robust in the long term)

We expect to receive your revised manuscript within two weeks. 

*Published Peer Review History*

*Early Version*

Sincerely,

Roli Roberts

Senior Editor,

rroberts@plos.org,

PLOS Biology

DATA NOT SHOWN?

REVIEWERS' COMMENTS:

Reviewer #2:

The authors thoroughly addressed all my comments and made the necessary adjustments in the text. I do not have any more concerns. This is very nice work.

Reviewer #3:

I am satisfied with how the authors addressed all my points. I will just mention a couple of details that can be corrected after acceptance:

I believe the G force values should be 1000-fold higher in the Methods Section "RNA extraction and sequencing".

Fig 4B mentioned in line 646 should be Fig 5B, while Fig 8B mentioned in line 666 should be Fig 8A.

This massive work is another great contribution of these authors to the field.

Reviewer #4:

[Accept - no comments]

---

## [Editor Report · Decision Letter 3]

20 Sep 2021

Dear Dr Hall,

On behalf of my colleagues and the Academic Editor, Arjan de Visser, I'm pleased to say that we can in principle offer to publish your Research Article "Plasmid fitness costs are caused by specific genetic conflicts enabling resolution by compensatory mutation" in PLOS Biology, provided you address any remaining formatting and reporting issues. These will be detailed in an email that will follow this letter and that you will usually receive within 2-3 business days, during which time no action is required from you. Please note that we will not be able to formally accept your manuscript and schedule it for publication until you have made the required changes.

In answer to your question about commissioning artwork, yes, you should be able to upload Striking Images after this point, but the practicalities of this are probably best discussed with the colleagues who will send you the aforementioned email in the next few days. If for some reason you miss the boat, feel free to email any images to me (rroberts@plos.org), as I'm involved in most of the aspects of the journal that would use such material (homepage, social media, etc.).

PRESS: We frequently collaborate with press offices. If your institution or institutions have a press office, please notify them about your upcoming paper at this point, to enable them to help maximise its impact. If the press office is planning to promote your findings, we would be grateful if they could coordinate with biologypress@plos.org. If you have not yet opted out of the early version process, we ask that you notify us immediately of any press plans so that we may do so on your behalf.

Sincerely, 

Roli Roberts

Roland G Roberts, PhD 

Senior Editor 

PLOS Biology

rroberts@plos.org